# Evaluation of traffic exhaust contributions to ambient carbonaceous submicron particulate matter in an urban roadside environment in Hong Kong

Berto Paul Lee[1], Peter Kwok Keung Louie[2], Connie Luk[2], and Chak Keung Chan[1]*

[1]School of Energy and Environment, City University of Hong Kong, Hong Kong, China

[2]Environmental Protection Department, HKSAR Government, Wan Chai, Hong Kong, China

*Correspondence to:* Chak K. Chan (chak.k.chan@cityu.edu.hk)

**Abstract.** Road traffic has significant impacts on air quality particularly in densely urbanized and populated areas where vehicle emissions are a major local source of ambient particulate matter. Engine type (i.e. fuel use) significantly impacts the chemical characteristics of tailpipe emission and thus the distribution of engine types in traffic impacts measured ambient concentrations. This study provides an estimation of the contribution of vehicles powered by different fuels (gasoline, diesel, LPG) to carbonaceous submicron aerosol ($PM_1$) based on ambient aerosol mass spectrometer (AMS) and elemental carbon (EC) measurements and vehicle count data in an urban inner city environment in Hong Kong with the aim to gauge the importance of different engine types to particulate matter burdens in a typical urban street canyon. On an average per-vehicle-basis, gasoline vehicles emitted 75% and 93% more organics than diesel and LPG vehicles respectively, while EC emissions from diesel vehicles were 45% higher than those from gasoline vehicles. LPG vehicles showed no appreciable contributions to EC and thus overall represented a small contributor to traffic-related primary ambient $PM_1$ despite their high abundance (~30%) in the traffic mix. Total carbonaceous particle mass contributions to ambient $PM_1$ from diesel engines were only marginally higher (~4%) than those from gasoline engines, which is likely an effect of recently introduced control strategies targeted at commercial vehicles and buses. Overall, gasoline vehicles contributed 1.2 µg/m³ of EC and 1.1 µg/m³ of organics, LPG vehicles 0.6 µg/m³ of organics and diesel vehicles 2.0 µg/m³ of EC and 0.7 µg/m³ of organics to ambient carbonaceous $PM_1$.

## 1. Introduction

### 1.1. Particulate matter from motor vehicles in urban areas

Emissions from on-road motor vehicles comprise gas-phase species (CO, $CO_2$, $NO_x$, $SO_2$) and volatile organic compounds among them important precursors of secondary organic aerosol (SOA), as well primary particulate matter predominantly in the fine ($PM_{2.5}$) and ultrafine particle size range ($PM_{0.1}$) mostly as carbonaceous species which encompass primary organic aerosol (POA) components and elemental carbon (Giechaskiel et al., 2014). Increased acute occurrence and risk of chronic development of cardiovascular and pulmonary diseases are important epidemiological effects of particulate matter inhalation (Davidson et al., 2005;Pope, 2007;Valavanidis et al., 2008;World Health Organization, 2011). In urban areas with high population and building densities, proximity to vehicle emissions poses a significant public health risk and renders paramount importance to the characterization and quantification of vehicle emissions (Kumar et al., 2014;Uherek et al., 2010).

Their contribution to total ambient concentrations however remains elusive with considerable variability in measured emission rates and species composition between as well as within vehicle classes from exhaust measurements in laboratory settings, vehicle chase or portable emission measurement systems (PEMS) studies (Franco et al., 2013;Kwak et al., 2014;Karjalainen et al., 2014;Giechaskiel et al., 2014;Alves et al., 2015) . These differences have been attributed to numerous influencing factors such as vehicle age, fuel use, operational parameters, environmental conditions and the subsequent introduction of more advanced engine technology (e.g. gasoline direct injection / GDI) and exhaust after-treatment (e.g. diesel particulate filters / DPF) (May et al., 2014). Apart from combustion-related compounds, species originating from lubricating oils have been identified as major components in primary gasoline and diesel exhaust and may be responsible for more than half of the total in-use POA emissions (Worton et al., 2014).

Post-emission oxidation of gas-phase organic constituents leads to atmospheric formation of secondary organic aerosol. Recent renewed interest in the SOA forming potential of motor vehicle exhaust through ambient observations, oxidation chamber and flow reactor studies (Ensberg et al., 2014;Gentner et al., 2012;Gentner et al., 2017;Platt et al., 2013;Zhao et al., 2015) affirms its potency in contributing significantly to urban SOA burdens however with still considerable quantitative uncertainties (Gentner et al., 2017).

Investigations on the impacts of motor vehicle emissions on ambient air pollution and related effects such as health implications (Kheirbek et al., 2016;Zhang and Batterman, 2013;Levy et al., 2010) typically rely on combinations of source inventories with dispersion modelling and combined physical-chemical models with significant uncertainties arising from the complexity of the urban built environment (Kumar et al., 2011;Belcher et al., 2013), inventory currency and accuracy of emission factors (Simon et al., 2008;Fuzzi et al., 2015). The latter is critically affected by the design of dynamometer test cycles with discrepancies having been noted between type approval procedure derived emissions and on-road measured driving emissions, which underlines the continued need to evaluate and monitor contributions of traffic emissions to urban air pollution by direct ambient measurements (Harrison and Beddows, 2017). A dominant factor that affects both gas- and particle-phase species distribution is the engine type used (Alves et al., 2015;Franco et al., 2013;Jang et al., 2016;Kam et al., 2012;Karjalainen et al.,

2014;Kwak et al., 2014), i.e. the distribution of engine types in the traffic mix will have significant influence on ambient concentrations. In the US gasoline fuel use in light duty vehicles is wide spread contributing to a greater dominance of gasoline-related VOC and particle-phase organic emissions, whereas in Europe diesel-vehicle use prevails leading to higher ambient burdens of black carbon (BC) (Zotter et al., 2014;Gentner et al., 2017). Evaluating the contribution of different vehicle groups to exhaust related species in ambient measurements is therefore of vital interest to gauge their current importance to urban air pollution and to acquire a baseline for assessing the efficacy of control policies and future effects of traffic mix and vehicle technology changes.

In this study, we evaluated the contributions of the three predominant engine-type vehicle groups (gasoline, diesel, LPG) in Hong Kong to primary carbonaceous aerosol by combining time-resolved ambient measurements by aerosol mass spectrometry (AMS) and ECOC analysis with vehicle count data. Characterization studies on road traffic emissions in Hong Kong are sparse due to the complexity of the urban built environment and the encountered transient engine loads which make emission factor and dispersion modeling approaches difficult to implement. Measurements were undertaken within a street canyon at a typical inner-city location where urban driving patterns with frequent stop-and-go traffic are prevalent, which may not be adequately reflected in dynamometer or cruising speed chase studies, but are more representative of pedestrian exposure levels particularly in view of growing concerns about exposure to air pollutants and their public health impacts in densely populated and built-up environments.

## 1.2. Background of vehicle emission studies in Hong Kong

Despite a low vehicle-to-population ratio (<10%), private and public road transport of passengers and goods is extensive throughout Hong Kong. In the year of this study (2013) 680,914 vehicles were registered in the Hong Kong Special Administrative Region (HKSAR) which accounted for an average of 35.17 million vehicle kilometers per day across all modes of road transport such as heavy goods haulage, commercial transport, private and public transportation. Passenger journeys on Hong Kong's three major franchised bus companies in 2013 were in excess of 1.1 billion (Transport Department, 2014). Road traffic has been recognized as a major local emission source and subjected to various regulatory efforts over the years, such as a large-scale fuel switch of taxis to a virtually all-LPG fleet, partial conversion of minibuses to LPG engines, ex-gratia incentive schemes for vehicle replacement or control device retrofitting and increased stringency in new-vehicle imports (Environmental Protection Department, 2013;Lyu et al., 2017;Ning et al., 2012).

Characterization studies on particle-phase species associated with traffic emissions in Hong Kong have mainly focused on semi-continuous online and offline filter analysis with carbonaceous components (sum of elemental carbon and organic matter) typically constituting the bulk (45-70%) of particulate mass in $PM_{2.5}$ at roadside sampling sites and up to 82% in road tunnel environments (Cheng et al., 2010;Huang et al., 2014;Lee et al., 2006;Louie et al., 2005;Research Centre of Environmental Technology and Management, 2005). Among major primary and secondary aerosol sources of ambient particulate matter which were identified from speciated long-term filter samples (1998-2008) of $PM_{2.5}$ by source apportionment analysis utilizing

positive matrix factorization, vehicle emission exhibited substantial contributions: at general measurement sites (urban, rooftop), vehicle emissions accounted for 8-9 µg m$^{-3}$ equivalent to 11-25% of total PM$_{2.5}$ with lower fractions in the fall and winter seasons primarily due to the masking influence of regional and long-range transport. Higher fractions (22-47%) and concentrations (19-23 µg m$^{-3}$) were resolved from roadside samples (street level) and with seasonal variations similar to those at the general sites attributed to similar effects of non-local pollution sources (Yuan et al., 2013).

Generally, low overall organic carbon (OC) to elemental carbon (EC) ratios (0.6-0.8) which are typical for locations in direct proximity to primary combustion sources were observed. In comparison, samples from urban rooftop sites exhibited lower contributions of carbonaceous constituents (<50%) and were impacted more by oxidized secondary organic species with correspondingly higher overall OC/EC ratios ~1.9 (Louie et al., 2005;Cheng et al., 2010;Lee et al., 2006). Impacts of long-range transport of continental aerosol were evident in both higher total PM and higher OC concentrations in the winter months. By contrast, seasonal variations of EC concentrations were weak. Higher EC concentrations were typically associated with rush hour traffic and confirm the local nature of EC and its relation to mainly on-road traffic (Lee et al., 2006;Cheng et al., 2010;Huang et al., 2014;Louie et al., 2005). While earlier studies (*samples dated pre-2005*) reported EC concentrations in the range of 11-25 µg C m$^{-3}$ in PM$_{2.5}$ at the open roadside (Louie et al., 2005;Cheng et al., 2010;Lee et al., 2006), more recent measurements (*2012-2013*) indicate a significant reduction in overall roadside EC burdens with concentrations ~ 5 µg C m$^{-3}$ (Huang et al., 2014;Sun et al., 2016;Lee et al., 2015). Vehicle-related OC and organic matter (OM) concentrations at a roadside measurement site were estimated to ~2 µg C m$^{-3}$ and ~7 µg m$^{-3}$ respectively in 2012, accounting for one-third of total OC in PM$_{2.5}$ and one-quarter of total PM$_{2.5}$ (Huang et al., 2014).

Contributions of traffic emissions to ambient gas-phase species were evaluated utilizing VOC canister samples collected in locations with different dominant-vehicle types in Hong Kong in 2003 (Ho et al., 2013). Propane, n- and i-butane accounted for over 80% of identified species in the LPG vehicle dominated samples, while toluene was the most abundant species in the gasoline vehicle dominated samples. Ethene and ethyne were enriched in diesel exhaust dominated samples. Contributions of vehicle-related VOCs to ozone formation were evaluated from the maximum incremental reactivity, with toluene (15-17%) and propene (7-8%) accounting for the largest shares (Ho et al., 2013).

Vehicle emission factors were estimated by a mass-balance model with differential measurements near the entrance and exit of a tunnel bore in 2003 and manual vehicle counts. Diesel vehicles were found to contribute substantially more to NO$_x$, PM$_{2.5,}$ and PAH than gasoline vehicles which dominated CO emissions. Diesel and LPG-fueled vehicles dominated total measured VOCs and total measured carbonyl compounds. EC emission factors of diesel vehicles (131 mg veh$^{-1}$ km$^{-1}$) were ~40 times higher than those of non-diesel (i.e. gasoline and LPG-fueled) vehicles (3.2 mg veh$^{-1}$ km$^{-1}$). For OC, the difference amounted to a smaller factor of ~8 with emission factors of 67.9 mg veh$^{-1}$ km$^{-1}$ and 8.5 mg veh$^{-1}$ km$^{-1}$ for diesel and non-diesel vehicles respectively. In terms of fractional contribution, OC made up 51% of PM$_{2.5}$ in non-diesel emissions, while EC made up 51% of PM$_{2.5}$ in diesel emissions. Contributions of inorganic species including ions and trace metals in traffic emissions were minor amounting to only ~10% (Cheng et al., 2010;Research Centre of Environmental Technology and Management, 2005). While these emission factors compared fairly well with those from similar studies undertaken in the 1990s (Gertler et al.,

2001;Lowenthal et al., 1994;Kirchstetter et al., 1999) in other parts of the world, they are unlikely to reflect the properties of
contemporary mobile road traffic emissions in Hong Kong with substantial changes in vehicle fleet composition and the
emergence of more advanced engine and emission control technologies over the last 15 years.
A mobile measurement platform with an array of on-board PM and gas monitors was deployed in early 2012 for a more up-
to-date characterization of fuel-based emission factors of PM, $NO_x$, and butane by chasing vehicle plumes on major roadways
and highways (Ning et al., 2012). The measurements focused on the evaluation of heavy duty diesel trucks (HDDV), franchised
diesel buses (DB) and LPG fueled light buses (LB) with plumes sampled at cruise speed to mitigate biases from engine load-
dependent emission variability. HDDVs were the highest emitters of BC (1.6 g $kg^{-1}$ of fuel) followed by diesel buses (1.1 g
$kg^{-1}$ of fuel) and LPG light buses (0.1 g $kg^{-1}$ of fuel). Analogously, average particle number emission factors were 3.1 x $10^{15}$
$kg^{-1}$ of fuel, 3.1 x $10^{15}$ $kg^{-1}$ of fuel and 0.9 x $10^{15}$ $kg^{-1}$ of fuel for HDDVs, diesel buses and LPG light buses respectively.
On-road remote sensing measurements of gas-phase emissions showed that gasoline vehicles had higher CO and HC emissions
and lower NO emissions compared to diesel vehicles, while LPG vehicles overall emitted more gaseous pollutants than their
light-duty petrol and diesel powered vehicle counterparts. (Lau et al., 2012). The higher LPG vehicle emission factors were
attributed to the higher usage rate of mostly LPG-powered taxis and light buses combined with insufficient vehicle
maintenance, imploring a gap between tightened regulations and effectiveness in the reduction of on-road emissions. Overall,
lower emission factors in newer vehicles within the same engine category were attributed to the efficacy of fitted emission
control technologies.
Despite road traffic representing a major local source of ambient particulate matter in Hong Kong, characterization studies on
vehicle emissions in Hong Kong remain sparse. While most previous studies evaluated traffic exhaust contributions in confined
environments (tunnels), focal locations (e.g. bus terminus, taxi stand) or by plume chase, this work provides a quantitative
estimation of vehicle exhaust related carbonaceous aerosol contributions and their relationship to traffic flow characteristics
from ambient measurements in a more complex urban environment. We aim to complement previous studies which only
estimated overall traffic-related aerosol contributions from online and offline ECOC measurements at different time resolutions
(from hourly to 24h samples) (Louie et al., 2005;Huang et al., 2014) and online mass spectrometric methods (Sun et al.,
2016;Lee et al., 2015) at the same location by combining resolved traffic-related organic aerosol concentrations from factor
analysis of aerosol mass spectrometer (AMS) data, online EC measurements and traffic count data to gain an improved
understanding of the influence of traffic composition on traffic-related submicron carbonaceous aerosol.

## 2. Methodology

Sampling of ambient particulate matter by a high-resolution time-of-flight aerosol mass spectrometer (HR-ToF-AMS, Aerodyne Inc.) took place in spring 2013 (7 March 2013 to 15 May 2013) and summer 2013 (16 May 2013 to 19 July 2013). Measurements were undertaken at a ground-level site on a pedestrian island at a road junction in the Mong Kok (MK) district of the densely populated and built-up Kowloon peninsula. Additional data such as meteorological data (wind, temperature, relative humidity, solar irradiation), concentrations of volatile organic compounds (VOCs), as well as trace gases ($NO_x$, $SO_2$, and $O_3$) were available from the adjacent air quality monitoring site (AQMS) operated by the Environmental Protection Department (EPD) of the HKSAR Government on the same road divider island. Details on employed instrument model and sampling methodologies are available in other dedicated publications (Lee et al., 2015;Environmental Protection Department, 2016;Sun et al., 2016). Elemental and organic carbon (EC, OC) were measured (semi-continuous thermo-optical ECOC analyzer RT-3131, Sunset Inc., USA) at the AQMS at 1h time resolution and at similar sampling height (~2m) employing a modified National Institute for Occupational Safety and Health (NIOSH) method 5040 protocol which has been described in a previous study (Huang et al., 2014). Concentrations of EC and OC in $PM_1$ were estimated by converting the raw $PM_{2.5}$ concentrations using a $PM_1$ to $PM_{2.5}$ ratio of 0.8 representing the overall average of $PM_1$ to $PM_{2.5}$ ratios from previous roadside sampling studies (0.75-0.85) in Hong Kong (Cheng et al., 2006;Lee et al., 2006).

AMS data were treated according to general AMS data treatment principles (DeCarlo et al., 2006;Jimenez et al., 2003) with standard software packages (SQUIRREL v1.53G, PIKA v1.12G). Analysis of the unit-mass resolution mass spectra yielded non-refractory submicron particle species concentrations of major inorganic constituents (SO4, NO3, NH4, Chl) and total organics at a base time resolution of 10 min. Positive Matrix Factorization (PMF) was used to deconvolute high-resolution organic mass spectra acquired at 10 min time resolution following recommended PMF guidelines for AMS data (Zhang et al., 2011) with the AMS PMF analysis toolkit (Ulbrich et al., 2009). Six organic aerosol (OA) factors were identified encompassing three secondary organic aerosol (SOA) and three primary organic aerosol (POA) factors of which one was attributed to traffic emissions (hydrocarbon-like organic aerosol, HOA (Zhang et al., 2005)) and two to cooking activities. OA factor concentrations were established using the fractional factor contributions from the high-resolution PMF analysis and the total submicron organics concentrations from the unit-mass resolution analysis. Both unit- and high-resolution mass spectral derived data were averaged to 1h time resolution to match ECOC measurements and traffic count data (*see below*), assessed from 28 May to 31 May, 2013. Average species concentrations measured during this time as well as the seasonal averages (spring, summer) from the whole measurement campaign are summarized in Table 1. Further specific details on instrument parameters, data analysis and an overview of the general characteristics of submicron particulate matter from the same campaign discussing the whole measurement period can be found in a previous publication (Lee et al., 2015).

Details on traffic composition are usually obtained by concurrent video image recordings and subsequent manual assessment of vehicle numbers and identification of vehicle types with classifications e.g. by vehicle function (Wang et al., 2016) or broader groups such as fuel types (Cui et al., 2016). In this work, vehicle data were obtained from the Hong Kong government,

which conducted a counting exercise during the sampling period over three weekdays (noon 28 May 2013 to noon 31 May
2013, Tue to Fri) using automated license plate recognition (ALPR) with four infrared cameras on-site. Traffic along Lai Chi
Kok road was monitored with three cameras monitoring all three lanes in one direction (out- or inbound) and the remaining
camera monitoring one lane of the opposite direction. Camera positions and the monitored single lane were swapped in 2-3h
intervals. Non-personalized data on registered vehicles were obtained, including license class (*vehicle type*), year of
manufacture (*vehicle age*) and engine type (*gasoline, diesel, LPG, others*), thus providing a direct linkage to actual vehicle
inventory data and circumventing the need for manual assessments. The obtained data were pooled by engine type (i.e. type
of fuel used) to construct an hourly time series of diesel, gasoline and LPG powered vehicles.
Measured HOA and EC during the traffic counting period were decomposed by multiple linear regression (MLR) based on the
time series of these engine count data, i.e. the hourly time series of HOA and EC were regarded as functions of the count of
diesel, gasoline and LPG powered vehicles representing the independent regression variables. To reduce the skewing impact
of scattered spikes in vehicle count number, HOA and EC concentrations, the time series were subjected to three-point box
smoothing prior to regression analysis. Multiple linear regression was performed for two and three factor solutions and with a
constrained zero intercept. A non-zero intercept may not be physically meaningful as it requires the assumption of a constant
background level of HOA and EC regardless of changes in surface wind, air mass influence or diurnal changes in background
source strength. Contrarily, traffic is largely homogenous in the vicinity of the sampling site (dense inner city traffic) and other
possible combustion sources were sufficiently removed (i.e. shipping with ~2km of straight-line distance to the nearest
coastline) and shielded by complex and tall urban geometry from the ground-level sampling site, i.e. impacts of transport on
measured traffic-related carbonaceous constituents can be assumed to be of minor influence. MLR solutions were assessed per
statistical significance of resolved regression coefficients, adjusted $R^2$ of the regression and the distribution of residuals.
Further details are discussed in the corresponding discussion section 3.3.1.

## 3. Results

### 3.1. Chemical and temporal characteristics of traffic-related carbonaceous species

Organic aerosol components were resolved by PMF from the organic part of the measured high-resolution mass spectra
acquired during the measurement campaign and scaled to the measured total organic species concentrations based on the unit-
mass resolution acquisition mode (Lee et al., 2015). Hydrocarbon-like organic aerosol (HOA) was resolved and identified as
the traffic exhaust related factor, exhibiting chemical characteristics typically associated with freshly emitted primary organic
aerosol from fossil fuel combustion (Zhang et al., 2005) and with appreciable correlations to other traffic-related species such
as $NO_x$ or pentanes (Lee et al., 2015). Its mass spectrum (Fig. S1a in the Supplement) is dominated by ions of saturated
hydrocarbons ($C_nH_{2n+1}$ series) and its elemental composition is largely void of oxygenated components with correspondingly
low O/C ratio of 0.066 and a high H/C ratio of 2.073. These findings are in line with most ambient studies that have identified
a similar factor, e.g. (Sun et al., 2012;Sun et al., 2011;Mohr et al., 2012;Lanz et al., 2007;Huang et al., 2010;Sun et al.,
2010;Aiken et al., 2009;Setyan et al., 2012).
The highest fractions of HOA among total organic particulate matter in $PM_1$ were observed with southeasterly and
northwesterly surface winds following the street canyon axis along Lai Chi Kok Road (Fig. S2a in the Supplement). Average
HOA mass concentrations varied between spring (3.5 µg m$^{-3}$) and summer (2.0 µg m$^{-3}$), with an overall campaign average of
2.8 µg m$^{-3}$. Diurnal variations of HOA were substantial, with mass concentrations rising sharply in the early morning (06:00),
increasing almost linearly during the morning rush hour at rates between 0.6 and 0.8 µg m$^{-3}$ h$^{-1}$ and leveling off beyond 09:00
(Fig. 1a). 90% of the daily HOA was accumulated between the start of the rush hour and midnight. Minimum HOA
concentrations in the low traffic period (00:00 to 06:00) amounted to ~1.7 µg m$^{-3}$ in spring and ~0.7 µg m$^{-3}$ in summer. Average
post rush-hour concentrations (09:00 to 00:00) were 4.1 µg m$^{-3}$ and 2.6 µg m$^{-3}$ in spring and summer respectively.
Elemental carbon (EC) and organic carbon (OC) in $PM_{2.5}$ were measured semi-continuously at the adjacent AQMS following
Huang et al. (Huang et al., 2014) and converted to $PM_1$ as described in the methodology section. In the following, we refer to
the $PM_1$ concentrations for discussion. Diurnal variations of EC were similar to those of HOA, however, nominal
concentrations were not affected by the seasonal transition. This led to significantly different EC/HOA ratios of 0.86 in spring
and 1.78 in summer (Fig. S1b in the Supplement). As the measurement location falls within the Kowloon urban area, which is
characterized by uniform annual traffic flows (Transport Department, 2014), the observed decrease in HOA concentrations
was more likely due to significant amounts of HOA being semi-volatile and partitioning in the gas-phase portion of the exhaust
emissions in summer than changes in traffic flows. This is also consistent with the trend in relative diurnal variation of HOA
(depicted cumulatively in Fig. 1b) which remained constant across seasons despite the seasonality in nominal concentrations,
and the overall negative correlation of HOA mass concentration and temperature (Fig. S2b in the Supplement). The semi-
volatility of HOA in this study is inferred from the observed trend in ambient mass concentrations and affirms more recent
studies on the partitioning behavior of POA components. Evaporation of semi-volatile components upon dilution decreases
ambient POA concentrations considerably and is expected to be temperature-dependent (Robinson et al., 2007), as is the case
in this study. Measurements of thermally denuded particles of primary origin by TD-AMS indicate that POA components can
exhibit properties of semi-volatility on a scale similar to SOA in urban environments (Huffman et al., 2009a;Huffman et al.,
2009b), and more recent studies suggest that HOA possesses a relatively wide volatility distribution with up to 50-60% of its
mass consisting of semi-volatile organic compounds (Cappa and Jimenez, 2010;Paciga et al., 2016). As noted earlier, in this
study the seasonal decrease in submicron particle-phase organic mass concentrations (HOA) by 40% was associated with a
substantial increase in mean temperature by 7°C between spring (23°C ± 3°C) and summer (30°C ± 2°C).
Both HOA and EC exhibited significant reductions in overall mass concentrations on Sundays between 14% and 27%,
compared to the rest of the week (Monday to Saturday) as shown in Fig. 1d. Days with less than 75% (>18h) data availability
were excluded from both time series in the analysis to obviate bias in the daily averaging from their strong diurnal patterns, as
opposed to the original data presented by Lee at al. (2015) where the entire unabridged time series were used. Mean reductions
in nominal mass concentrations were generally quite similar in the range of 0.8 – 1.0 µg m$^{-3}$ for EC and ~0.5 µg m$^{-3}$ for HOA.

The greater overall reduction in EC indicates a greater reduction in the number of diesel vehicles on Sundays, owing to reduced commercial traffic and lower bus frequency. Contributions of vehicle emissions to ambient organic aerosol have been derived from one-year long measurements at the same site based on analysis of semi-continuous ECOC measurements (Huang et al., 2014). The EC tracer method (Turpin and Huntzicker, 1991;Lim and Turpin, 2002) was employed which evaluates minimum OC/EC ratios, and OC and EC measurements during time periods of high primary source strength and low photochemical activities when OC is mainly of local origin and attributable to the dominant primary sources in the vicinity. A characteristic vehicle-related OC/EC value (OC/EC$_{vehicle}$) of 0.5 was derived from the 5% lowest summer OC/EC ratio data to approximate organic vehicular emissions. The reported monthly average primary organic matter concentrations from traffic in PM$_{2.5}$ (using an OM/OC conversion factor of 1.4) ranged between 2.7 – 3.4 µg m$^{-3}$ with little seasonal variation. To compare to our measurement base of PM$_1$, we use the previously mentioned average ratio of PM$_1$/PM$_{2.5}$ of 0.8 for conversion. The OC/EC approach would thus yield an approximate 2.2 – 2.7 µg m$^{-3}$ of traffic-related primary organics in PM$_1$ which compares well with the overall average concentration of traffic-related HOA (2.8 µg m$^{-3}$) covering the entire AMS measurement campaign (March – July 2013). However, we observed significantly different concentrations depending on the season with much higher concentrations (3.5 µg m$^{-3}$) in spring than in summer (2.0 µg m$^{-3}$). The lack of a clear seasonal variation in OC/EC derived traffic-related organics is due to the use of a fixed conversion factor directly dependent on ambient EC concentration which did not exhibit a strong seasonal behavior and thus could not capture the same trends observed from the AMS measurements. The derivation of the OC/EC$_{vehicle}$ ratio is subject to further uncertainty due to the substantial influence of cooking-related organics at the site (Lee et al., 2015;Sun et al., 2016), i.e. traffic emissions may not be the dominant source of "local" organics at the site (Table 2), as well as the influence of background SOA which is likely to persist even in time periods of low OC/EC ratios.

Nonetheless, the agreement of the whole-campaign average HOA concentrations (this study) and OC/EC$_{vehicle}$ derived traffic organics mass concentration (Huang et al., 2014) suggests that the long-term average primary organic aerosol contribution from vehicles is reasonably well-approximated with the OC/EC approach but may lead to over- and underestimation of actual organic concentrations in shorter time frames, e.g. different seasons and different times of the day.

## 3.2. Traffic and vehicle flow characteristics

During the three-day counting exercise, a total of 21,048 vehicles were registered (~7016 vehicles per day). This accounts for approximately 18% of the annual average daily traffic (A.A.D.T.) of 38,220 vehicles estimated in the 2013 Annual Traffic Census (Transport Department, 2014). The diurnal variation of the counted vehicle number, resolved by broad vehicle classes, and their varying contribution to the vehicle mix during the day are depicted in Fig. S2c in the Supplement. The maximum hourly vehicle number occurred towards the end of the rush-hour (~9:00) for most vehicle classes (except "*other vehicles*", which mostly comprised motorcycles). For the evening rush-hour (17:00-19:00), only the private car category exhibited a peak in vehicle number. Overall, the post rush-hour total vehicle count remained stable throughout the day, and only fell off after

19:00, largely due to a strong decrease in the number of goods and private vehicles. The variation in the fractional contribution of different vehicles types over the day resembles that assessed by the Transport Department (Transport Department, 2014) at the closest core traffic counting station at Nathan Road (Fig. S2c in the Supplement) and affirms that the measured traffic composition during the 72h counting period at Mong Kok was largely representative. At the Mong Kok site, smaller vehicles (*cars and taxis*) accounted for about 60% of total registered vehicles, while heavier vehicles (*buses, vans, trucks*) made up most of the remainder.

Total HOA concentrations showed good correlations ($R_{pr}$>0.6) with most of the measured vehicle-related VOCs, $NO_x$ and EC concentrations as well as the total number of passing vehicles during the counting period (Fig. 2). As with the overall campaign data, CO did not correlate with either HOA or EC, indicating the impact of significant non-traffic-related CO sources in the Mong Kok area. Among the counted buses, light buses and goods vehicle, over 96% were running on diesel engines, while 99.9% of counted taxis utilized LPG engines. Private vehicles were almost exclusively powered by gasoline engines (99.4%) due to government restrictions on the import and use of diesel-powered private vehicles at the time of this study.

### 3.3. Estimation of engine-type contributions

### 3.3.1.    Multiple Linear Regression of HOA and EC

Utilizing the detailed information on average daily traffic composition at the measurement site from the counting exercise, contributions of exhaust from different engine types to overall ambient HOA and EC concentrations can be evaluated. Multiple linear regression was carried out with the concentration time series of HOA and EC during the traffic counting period as the dependent variables, the pooled sum of counted diesel, gasoline and LPG vehicles as the independent variables, and a constrained zero intercept, assuming negligible transport of HOA and EC (*see Methodology section*).

Two factor solutions (diesel + gasoline), as well as three factor solutions (diesel + gasoline + LPG), were considered, as LPG vehicles are expected to contribute less to particle-phase and more to gas-phase emissions (Faiz et al., 1996). The key statistical output parameters (adjusted $R^2$, regression coefficients, p-values) of the multiple linear regression analysis for HOA and EC for two and three factor models are presented in Table S1 in the Supplement. For HOA, both two and three factor models yielded acceptable results with all resolved coefficients statistically significant at the 95% confidence level (p<0.05). Exclusion of LPG vehicles (two factor model) led to additional apportionment of HOA to both gasoline (+25%) and diesel (+18%) vehicles.  Overall model performance improved only slightly when moving from the two factor solution ($R^2_{adj}$=0.86) to the three factor solution ($R^2_{adj}$=0.90). Comparative studies on particulate vehicle emissions have reported non-negligible particle mass contributions of LPG vehicles. Estimated particle mass emission factors from LPG passenger vehicles were similar (Chan et al., 2007) or ~30% lower (Ristovski et al., 2005) compared to gasoline powered models in chassis dynamometer runs. In on-road environments, absolute particle number concentrations in bus exhaust plumes were lower in samples from CNG and LPG fueled buses compared to diesel and DME (dimethyl ether) fueled buses, but still 10 times higher than typical ambient background particle concentrations (Kwak et al., 2014). Evaluating HOA concentrations measured between 0:00 and 5:00

when LPG vehicles dominated the total vehicle population (45-60% of total counted vehicles) in our study shows a clear positive relationship between ambient HOA and LPG vehicle count (Fig. S3a in the Supplement). The two factor MLR model tends to underestimate measured HOA mass concentrations with the distribution of absolute residuals (*difference of reconstructed and measured HOA mass concentration*) biased to negative values whereas the three factor model yielded more normally distributed residuals (Fig. S3b, c in the Supplement). We thus consider the three factor solution including LPG vehicles as a more appropriate representation for the deconvolution of the ambient HOA mix from our measurements. For EC, a three factor representation could not resolve variable coefficients that were all statistically significant at the 95% confidence level yielding $p>0.05$ for the gasoline vehicle factor. While both diesel and gasoline vehicle number correlated appreciably with EC (Fig. S4c in the Supplement), the number of LPG vehicles lacked a corresponding relation and stayed almost constant over the range of measured EC concentrations, rendering the two factor model the more appropriate representation in deconvoluting measured ambient EC.

As discussed in the methodology section, three point box smoothing was applied to the vehicle counts and carbonaceous $PM_1$ species concentrations to reduce biasing of resolved regression parameters induced from spikes in the time series (Figure S5 in the Supplement). Comparing MLR results from the smoothed and raw time series data (Table S2 in the Supplement), impacts on the variable coefficients and their variability (standard deviations) were minor and factor statistical significance was not detrimentally affected. The smoothed data set overall yielded better results with higher adjusted $R^2$ as well as lower average and summed residual values. The smoothing process thus did not incur substantial additional uncertainties in the deconvolution of HOA and EC.

**3.3.2. Analysis of engine-type reconstructed carbonaceous components**

The time series of measured and reconstructed HOA and EC concentrations based on the regression coefficients are depicted individually in Fig. S6a and S6c in the Supplement and in combination in Fig. 3 to represent the sum of motor vehicle related primary carbonaceous particulate compounds, not including possible additional SOA species formed through subsequent atmospheric processing of gas- and particle-phase species in tailpipe exhaust. Gasoline vehicles accounted for more than 45% of total HOA, while diesel and LPG-fueled vehicles were responsible for similar shares of 25-29% (Fig. S6b in the Supplement). Over 60% of total EC in $PM_1$ was accounted for by diesel vehicles (Fig. S6d in the Supplement). The sum of HOA and EC in $PM_1$ represent approximately the total submicron primary carbonaceous aerosol contributions from road traffic. Despite the limited time frame of the counting exercise, the MLR factors adequately resolved the overall trend in EC and HOA concentration (Fig. 3a) and diurnal variation (Fig. 3b). Relative residuals from the reconstructed sum of HOA and EC were ≤25% with overestimation (*Residuals < 0*) predominant in the later afternoon hours between 14:00 and 18:00, while significant underestimation (*Residuals > 0*) was observed mostly in the late evening hours between 19:00 and 0:00. The semi-volatile character of HOA would contribute to the observed reduced afternoon particle-phase concentration and conversely enhanced nighttime particle-phase concentrations. Averaged over the whole 72h period, measured and reconstructed

carbonaceous submicron $PM_1$ differed by only 4% indicating that the overall reconstruction was not affected considerably by these diurnal trends in residuals. Vehicle emission factors are typically based on the concentration of emitted species per unit of consumed fuel (derived from $CO_2$ measurements) or per driven distance. As concurrent $CO_2$ measurement data were not available, emission factors were not evaluated in this study. Instead, the relative emission behavior of different vehicle groups and their impacts on ambient PM concentrations were approximated by normalizing the resolved engine-specific carbonaceous primary aerosol concentrations by the total number of vehicles in each category (Fig. 3b). On a per-vehicle basis (Fig. 3b), gasoline vehicles emitted 75% more HOA than diesel vehicles and 93% more HOA than LPG vehicles, while EC emissions from diesel vehicles were 45% higher than those from gasoline vehicles. Therefore, on average gasoline and diesel vehicles emitted similar total amounts of primary carbonaceous $PM_1$ and each about three times more than LPG powered vehicles. In terms of ambient concentrations, gasoline vehicles accounted for 78% more ambient HOA than diesel vehicles and 56% more ambient HOA than LPG-fueled vehicles. As opposed diesel vehicles accounted for 62% more ambient EC than gasoline-powered vehicles. Each diesel and gasoline vehicles contributed 75-85% more carbonaceous $PM_1$ mass than LPG vehicles, which despite making up ~30% of total counted traffic accounted for less than 13% of traffic-related primary $PM_1$ (Fig. 3e). It must be noted that these data represent the total average of the encountered diesel, gasoline and LPG vehicle fleet, notwithstanding that each of these broad classes is inhomogeneous consisting of a wide variety of vehicles of different size, engine power and age, and thus different individual emission characteristics.

As the fractional contributions of diesel and gasoline vehicles to ambient HOA and EC varied substantially, i.e. diesel vehicles emitted almost three times as much EC as HOA (EC/HOA=2.85) and gasoline vehicles contributed similar mass concentrations to both HOA and EC (EC/HOA=1.02). This caused significant diurnal variations in the EC/HOA ratio (Fig. S1c in the Supplement) with low values in the nighttime hours, where LPG- and gasoline-fueled vehicles were more dominant and highest values occurring at the tail of the morning rush-hour where fractions of goods vehicles and buses among overall traffic were highest. Previous measurements of vehicle emissions in a highway road tunnel in the San Francisco Bay Area (Dallmann et al., 2014) employing a soot-particle aerosol mass spectrometer (SP-AMS) yielded a similar characteristic ratio of black carbon to organic aerosol (BC/OA~2.6) for diesel truck plumes and a smaller ratio of BC/OA<0.1 for gasoline vehicle plumes, while filter samples at the same location from light duty vehicles location exhibited BC/OA ratios of ~0.7 for light-duty vehicles and ~2.1 for medium and heavy diesel trucks (Ban-Weiss et al., 2008). Dynamometer studies mostly reported EC/OC values for gasoline powered vehicles < 1 (Alves et al., 2015;Geller et al., 2006) indicating overall a greater importance of organics in particulate emissions.

The relatively high fraction of EC in gasoline related PM in our work likely stems from a combination of factors. HOA mass concentrations are derived from AMS measurements which are subject to limitations of inlet lens transmission, i.e. a standard lens as used in this study is expected to transmit efficiently (~100%) between 90nm and 700nm $D_{va}$ (Williams et al., 2013), and thus may cause a low bias in measured ambient HOA concentrations. However, these inlet losses are expected to only have limited effects on total measured submicron particle mass concentrations as mass- and volume-based particle concentrations from vehicle exhaust typically peak in the region of ~ 100 – 300nm (Fushimi et al., 2016;Ban-Weiss et al.,

2010). The observed seasonal variability of particle-phase organic concentrations due to partitioning of semi-volatile components into the gas-phase would also decrease particle-phase HOA concentrations in warmer seasons, thus elevating the observed ambient EC/HOA ratio with our measurements taking place in summer. While this would affect all engine types, the extent may vary with the distribution of semi-volatile species in the respective tailpipe emissions of different vehicle groups. Due to the characteristics of the sampling site (inner-city urban road traffic and proximity to both a road junction and pedestrian crossing) sampled emissions include a considerable fraction of variable and higher engine loads during the acceleration phase whereas road tunnel environments are characterized by largely constant engine loads and traveling speeds. As gasoline vehicles are typically not equipped with particle filters, they have been shown to emit significant amounts of EC under unstable engine loads (Karjalainen et al., 2014).

It has also been noted that changes in engine technology, i.e. the move from port fuel injection (PFI) to gasoline direct injection (GDI), may shift gasoline vehicle exhaust characteristics in favor of elemental carbon. Higher particulate matter mass emissions by a factor of 2 from GDI vehicles compared to PFI vehicles have been reported, which were mainly due to enhancements in EC emissions (Saliba et al., 2017). Similar observations were made in comparisons of PFI and direct injection spark ignition (DISI – a derivative of GDI) vehicles over both cold and hot-start conditions with higher total carbon (TC) emissions and higher EC/TC ratios for the DISI vehicles (Fushimi et al., 2016).

At the same time, various control schemes targeting diesel vehicle emissions have been introduced in recent years. In Hong Kong these included inter alia an incentive scheme in 2010 for the replacement of diesel commercial vehicles, the retrofitting of diesel particulate filters (DPF) on pre-Euro IV buses and subsequent replacement of older buses with Euro IV, V and VI standard models (Environment Bureau, 2013;Environmental Protection Department, 2013). $PM_{2.5}$ emission factors determined from dynamometer test runs of various vehicle types in Hong Kong were generally higher for diesel vehicles compared to gasoline and LPG vehicles (Chan et al., 2007), while we observed similar total particle mass emission on a per-vehicle basis for diesel and gasoline vehicles in this study. The dynamometer tests were conducted on relatively old vehicles (*newest date of manufacture: 2001*) and are unlikely to adequately reflect the ambient vehicle mix in our study, particularly with regard to the implementation of recent emission control programs. New engine technologies for diesel vehicles, such as DPFs, have been shown to greatly reduce both EC and OC emissions (Alves et al., 2015) thus leading to overall little total particle mass emissions (Li et al., 2014;Quiros et al., 2015) partially due to shifts in the particle size distributions towards a greater fraction of particles in the ultrafine mode (Giechaskiel et al., 2012). The Euro III, IV and V standards for trucks and buses were introduced in late 2000, late 2005 and late 2008 respectively. While the year of manufacture does not directly infer compliance to a specific emission standard, an approximation of the fraction of vehicles fulfilling a certain standard can be made by assuming that vehicles produced between 2001 and 2005, between 2006 and 2008 and between 2009 and 2013 very likely comply with the Euro III, IV and V standards respectively. In this case, it is assumed that emission standards were immediately or had already been adopted in vehicle models in the corresponding year of manufacture. Figure S7 in the Supplement depicts the number and proportion of vehicles of certain years of manufacture and their assumed Euro standard. For goods vehicles, 52% of counted vehicles were built between 2005 and 2013 (i.e. likely fulfilling Euro IV and V), while for buses the proportion

was slightly lower at 33%. With these two vehicle groups representing the bulk of diesel powered vehicles, an estimated 40% of diesel vehicles complied with Euro IV and Euro V standards during the time of our ambient measurements, further rationalizing the relatively low per-vehicle contribution of diesel vehicles to ambient exhaust-derived carbonaceous $PM_1$ in this study.

The specific characteristics of the Mong Kok measurement site as well as the continual changes in vehicle technology are not represented well in typical dynamometer driving cycles and tunnel studies and are at the least partially responsible for the observed differences to other studies reporting relative abundances of elemental carbon to organic mass concentrations.

## 4. Conclusion

Primary submicron carbonaceous aerosol was measured by combined HR-ToF-AMS and ECOC measurements, supplemented by various VOCs and other gas-phase species, at an inner-city urban roadside sampling site in the Mong Kok District in Hong Kong. Organic species concentrations (HOA) were of similar magnitude as traffic-related organic matter estimated previously, but with strong seasonal dependency with on average 40% lower concentrations in summer (2.0 µg/m$^3$) compared to spring (3.5 µg/m$^3$) likely due to greater fractions of semi-volatile species remaining in the gas-phase in the warmer season. EC concentrations remained at similar levels throughout the sampling period, leading to significantly different average EC/HOA ratios in spring (0.86) and summer (1.78). Sunday concentrations of both HOA and EC were 14-27% lower than during the rest of the week (Mon-Sat) reflecting significant reductions in traffic volume on Sundays. Larger relative reductions in EC compared to HOA indicate a greater reduction in the number of diesel vehicles, e.g. commercial trucks and buses.

Detailed vehicle counting data acquired during a 72h counting exercise in late May 2013 were utilized to decompose HOA and EC concentrations into vehicle-type factors (gasoline-, diesel- and LPG-powered vehicles) by multiple linear regression analysis. Gasoline vehicles contributed similar amounts of EC and HOA, while diesel vehicles emitted predominantly EC and to a lesser extent HOA. On an average per-vehicle basis, contributions of diesel and gasoline vehicles to carbonaceous $PM_1$ were similar, contrary to previous studies which attributed higher particulate matter emissions to diesel vehicles. This clear reduction is likely due to recent control strategies targeted at commercial vehicles and buses, which represent the bulk of diesel powered vehicles at the measurement site. This becomes especially important in view of the growing number of higher Euro standard diesel vehicles which, while effectively reducing overall emitted particle mass may emit similar number concentrations of particles with particle sizes having been shown to shift more into the ultrafine region (Tartakovsky et al., 2015). The number of LPG-vehicles did not exhibit a significant correlation with EC concentrations and was thus considered to be a negligible EC source. Its contributions to ambient primary traffic-related $PM_1$ were small (13%) despite representing about one-third of total counted vehicles.

While this study has focused on primary emissions, gas-phase emissions from the tailpipe can lead to subsequent condensation of organics or atmospheric oxidation to form secondary organic aerosol, significantly enhancing post-emission particulate matter concentrations (Gentner et al., 2012;Platt et al., 2013) and should be subject to further future investigation.

**Acknowledgments**

This work was supported by the Environment and Conservation Fund of Hong Kong (project number ECWW09EG04). Chak K. Chan gratefully acknowledges the startup fund of the City University of Hong Kong.

**Disclaimer**

The opinions expressed in this paper are those of the author and do not necessarily reflect the views or policies of the Government of the Hong Kong Special Administrative Region, nor does mention of trade names or commercial products constitute an endorsement or recommendation of their use.

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

**Tables**
**Table 1. S**pecies concentrations from AMS and total $PM_{2.5}$ mass from TEOM in (a); EC, OC and organic aerosol constituent
concentrations (AMS PMF analysis) in (b); average concentrations during the traffic counting period (May 28-31, 2013)
and seasonal averages (Spring: Mar – May 2013, Summer: May – Jul 2013 ) based on Lee at al. (2015) at the Mong Kok
site

| (a) | **μg/m³** Mean ± *SD* | **Org** (AMS) | **SO4** (AMS) | **NO3** (AMS) | **NH4** (AMS) | **Chl** (AMS) | **NR-PM₁** (AMS) | **PM₂.₅** (TEOM) |
|---|---|---|---|---|---|---|---|---|
| | May 28-31 | 5.9±3.0 | 4.8±1.2 | 0.3± 0.1 | 1.5±0.3 | <0.1 | 12.5±3.7 | 16.0±5.3 |
| | Spring | 12.8±7.6 | 7.0±3.8 | 2.5±2.1 | 2.5±1.5 | 0.4±0.3 | 25.3±13.1 | 32.3±12.4 |
| | Summer | 7.9±5.4 | 3.4±2.0 | 0.4±0.4 | 1.1±0.5 | <0.1 | 12.7±3.6 | 17.3±7.5 |


| (b) | **μg/m³** Mean ± *SD* | **EC** (PM₂.₅) | **OC** (PM₂.₅) | **EC** (PM₁) | **OC** (PM₁) | **HOA** (PM₁) | **COA[a]** (PM₁) | **SOA[b]** (PM₁) |
|---|---|---|---|---|---|---|---|---|
| | May 28-31 | 4.2±2.3 | 3.0±1.2 | 3.2±1.7 | 2.3±0.9 | 2.4±1.4 | 2.0±1.2 | 1.6±1.1 |
| | Spring | 4.3±2.6 | 7.6±3.9 | 3.2±1.9 | 5.7±2.9 | 3.5±2.4 | 4.4±4.3 | 4.9±3.4 |
| | Summer | 4.3±2.5 | 4.1±2.1 | 3.2±1.8 | 3.0±1.6 | 2.0±1.3 | 3.6±3.4 | 2.2±2.2 |

*Notes:* [a] Sum of two cooking-related PMF factors
[b] Sum of three SOA-related PMF factors


    **Figures**

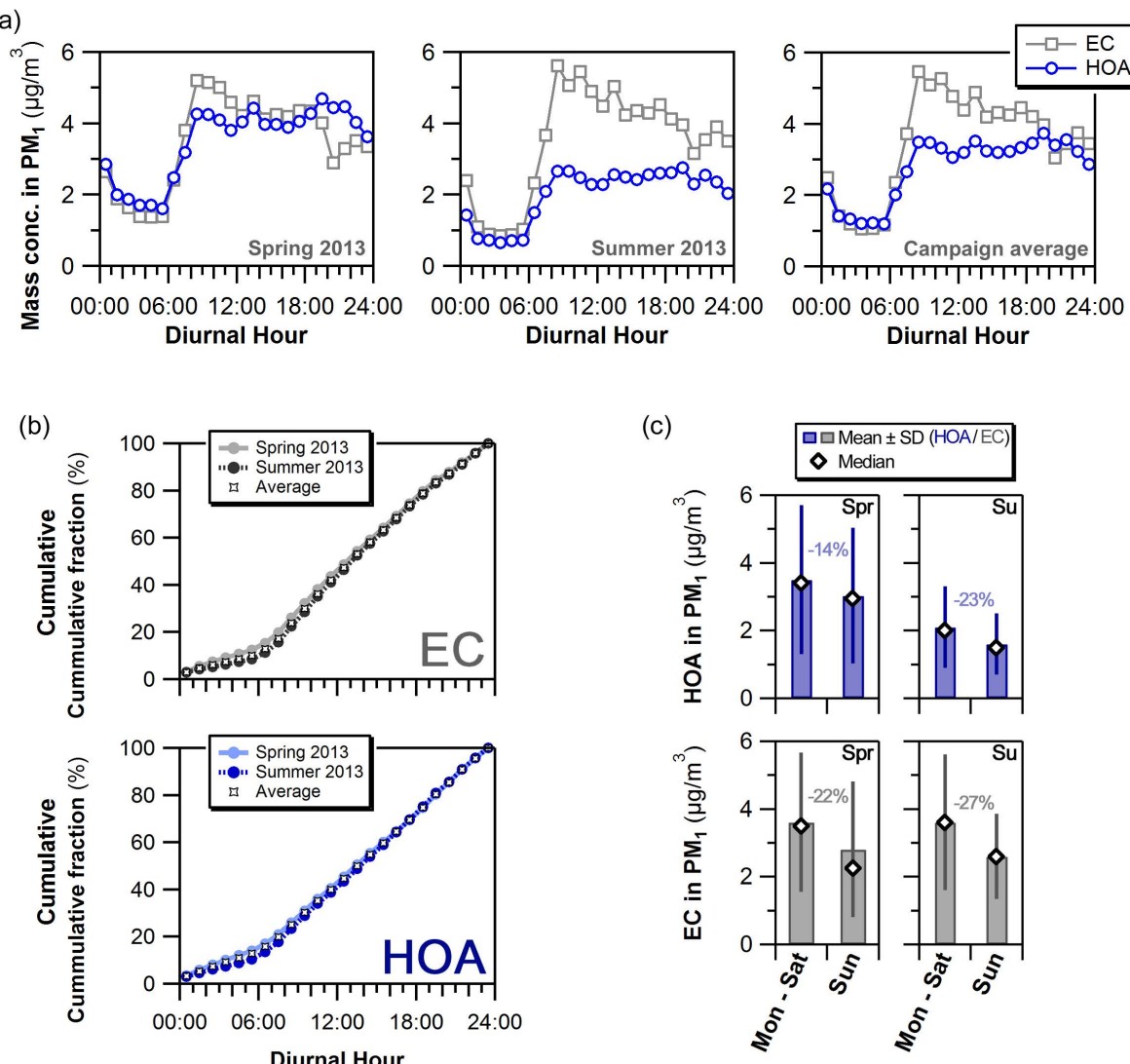


**Figure 1.**

(a) Diurnal variation of HOA and EC mass concentrations in spring 2013 (7 Mar 2013 to 15 May 2013), summer 2013 (16
May 2013 to 19 Jul 2013) and averaged over the entire sampling campaign (7 Mar 2013 to 19 Jul 2013)
(b) Diurnal variation of the cumulative fraction of total daily HOA and EC mass concentrations in spring 2013, summer
2013 and averaged over the entire sampling campaign
(c) Mean HOA and EC concentrations (*bar with standard deviations, open marker depicts median*) from Monday through
Saturday and on Sunday in spring 2013 (Spr) and summer 2013 (Su); days with >25% (>6h) of missing data were
excluded

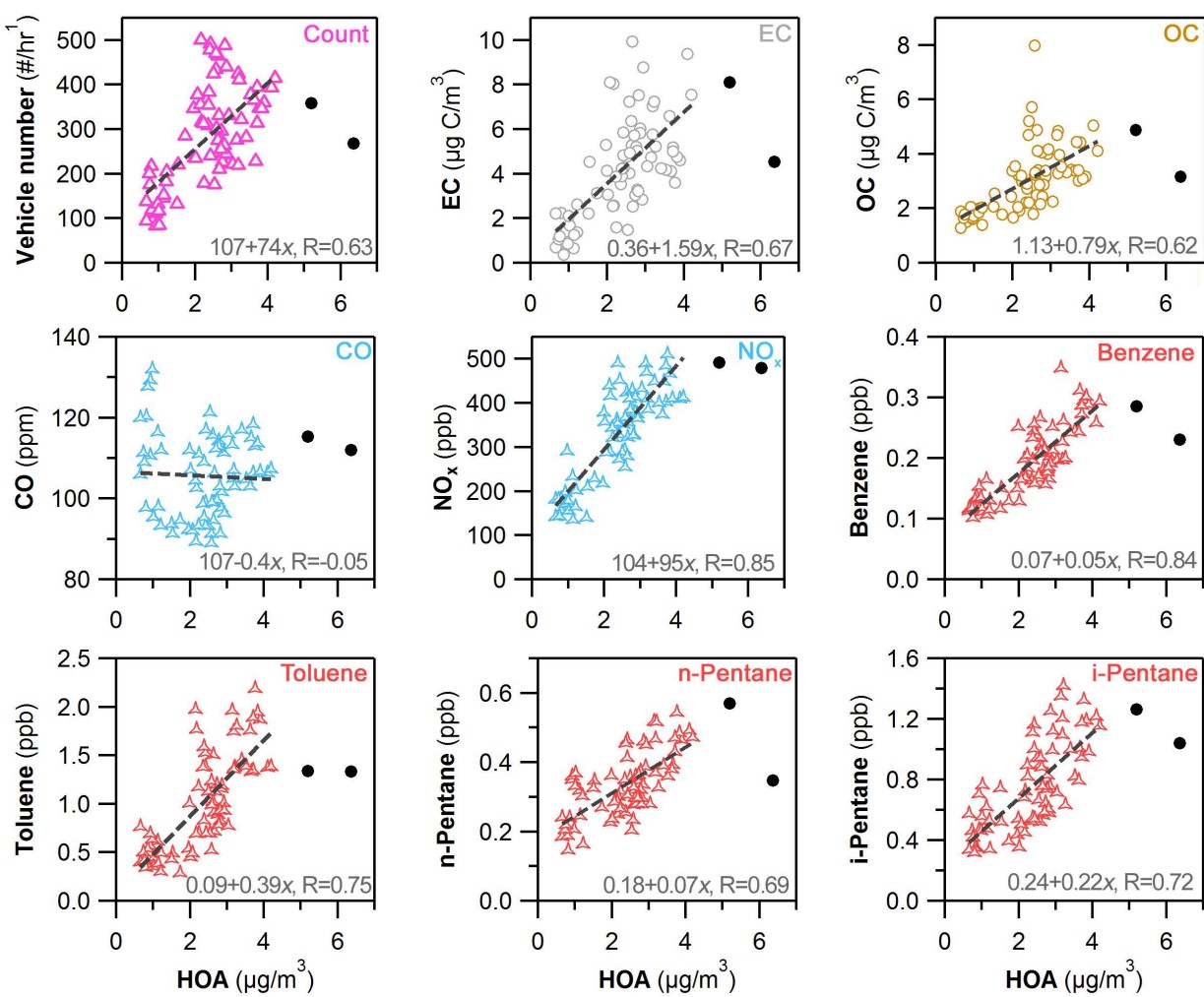


**Figure 2.**

Scatter plot of vehicle count, mass concentration of EC, OC and gas-phase concentrations of various VOCs, $NO_x$, and CO against HOA during 28-31 May 2013, *linear least squares fits with equations and Pearson's R values in grey, black dots signify outlier values excluded from the line fits*

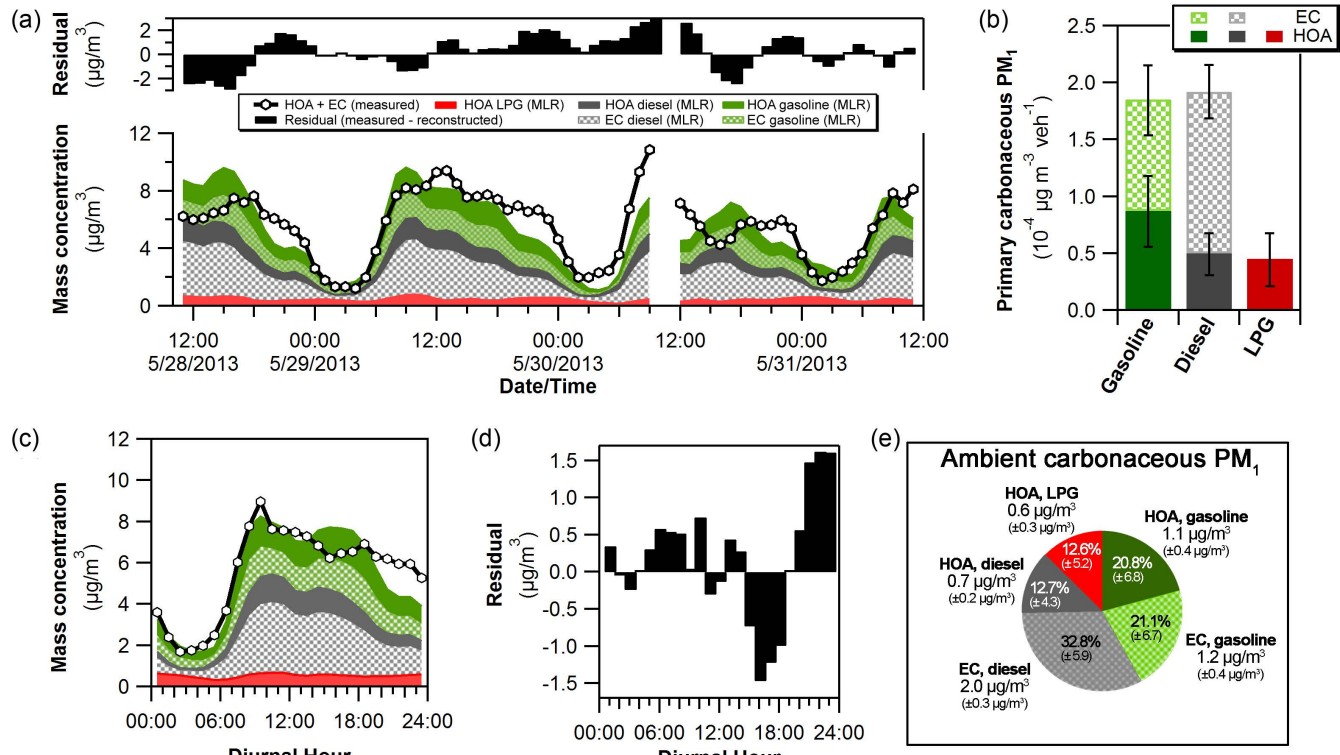

**Figure 3.**
(a) Time series of engine-type separated HOA and EC in PM₁, sum of actual measured HOA and EC in PM₁, as well as

residuals as the difference between resolved HOA+EC and measured HOA+EC

(b) Contribution of engine-type separated HOA and EC to PM₁ normalized by the total number of vehicles per engine category;

vehicle composition based on traffic count (28 – 31 May, 2013) and vehicle number scaled to average annual daily traffic

(Transport Department, 2014), uncertainties from resolved fit parameters (MLR) included

(c) Diurnal variation of vehicle-type separated HOA and EC concentrations as well as sum of actual measured HOA and EC

in PM₁

(d) Diurnal variation of residuals as the difference between resolved HOA+EC and measured HOA+EC in PM₁
(e) Average contribution of engine-type separated HOA and EC in total reconstructed HOA+EC in PM₁, uncertainties from

resolved fit parameters (MLR) included