# Peer review of "Evaluation of traffic exhaust contributions to ambient carbonaceous submicron particulate matter in an urban roadside environment in Hong Kong"

_Atmospheric Chemistry and Physics, 2017_

## Referee Comment (RC1) · Anonymous Referee #1 · 24 Jul 2017

GENERAL

This paper reports measurements of carbonaceous particles at a busy street canyon. The authors focus on primary hydrocarbon like (HOA) and elemental carbon (EC) fraction of the aerosol. They combine the concentration data with engine-type specific vehicle counts to obtain the contribution of diesel, gasoline, and LPG vehicles. The changes in emission regulations and the rapid development in engine and after-treatment techniques make efforts such as this necessary in evaluating the effect of the policies and development on the air quality. The measurement site is ideal for this kind of study because of the high contribution of traffic and the local measures taken

for decreasing the traffic emissions. The paper brings new information on an important question and merits publication, but there are some issues that should be treasted, as discussed below.

I understand that there are so many aspect on carbonaceous aerosols that the discussion needs to be limited somehow. However, I find the approach especially in the introduction too limited to the Hong Kong primary aerosol case. The introduction should shortly discuss the relative importance of the primary and secondary aerosol in urban settings and not just mention the secondary aspect in the last sentence of the paper. The authors should make a case why it is still important to study the primary components. In presenting and discussing the results, the authors should clearly, preferably in a table, report the primary and overall concentrations. The effect of renewing fuels and vehicle fleet on the urban concentrations has been studied also outside Hong Kong. These studies should be discussed, starting with the Harrison and Beddows, Nature 2017 paper.

There are recent papers on the same measurement cite, partly by the same authors. The authors should clearly state what is new here, especially compared to the Lee at al. 2015 JGRA and Huang et al ACP 2014 papers. There is an important methodological difference to the Huang paper in treating the traffic related OC. The difference in methodology and results, as well as the meaning of those should be discussed more explicitly.

Finally, the methodology and the analyses appear solid but they are treated so concisely that the reader needs to deduce what data was used. The methods are partly the same as used in the papers mentioned above, but this paper should also be readable separately. As this media allows some volume, the methodology should be described in more detail.

SPECIFIC

Abstract:

The abstract should highlight the new findings of this paper and report them quantitatively.

Methodology:

The methodology part should be rewritten to work on its own for this paper. Reading just this paper, it remains unclear what instrument data was used for, e.g., the engine specific contributions. Was this done on hourly basis as is indirectly indicated on the vehicle flow chapter? Obviously AMS data was used for the HOA, but what about EC: Aethalometer or the hourly EC measurement? And how did the latter two compare? Huang et al., 2014 should be cited already in the methodology part for the hourly EC/OC. An obvious measurement missing from the study is $CO_2$. This inhibits the calculation of emission factors and should be clearly stated.

Vehicle identification method to gain engine-type specific emissions seems to have been used at least in tunnels, although not necessarily documented very well. The authors should cite those, such as Cui et al., STOTEN 2016. They might also discuss the individual-vehicle specific approach of Wang et al., AtmEnv 2016.

Results:

The paper reports the contributions of the vehicle types, but also PM2.5, PM1, EC and OC concentrations should be given, maybe in a table gathering the results. Those reported earlier with a citation.

At least Zhang et al., ACP 2005 should be cited for using HOA as a surrogate for combustion POA. In discussing the seasonal variations and gaseous/particulate phase partitioning, Robinson et al., 2007 should be cited, maybe together with comparison to some other studies, such as Budisulistrioni et al., ACP 2016; Huffmann et al., ACP 2009.

The paper reports exceptionally high EC fractions (EC/HOA ratios) for gasoline vehicles. This is an interesting and potentially important finding, but also subject to controversy. While the site-specific driving patterns could explain some of this, other aspects should also be discussed. A high fraction of gasoline direct injection (GDI, DISI) vehicles could possibly affect this, as they have been found to exhibit high primary EC fractions (e.g. Karjalainen et al., ACP 2016; Fushini et al., AtmEnv 2016). On the other hand, the HOA concentration of this study comes from AMS, while the EC does not. The low sampling efficiency of the AMS for sub-50 nm particles could cause low HOA concentration as a measurement artefact, especially if a high mass fraction of the POA is within the nucleation mode.

Details

In section 3.3.2. it would be good to explicitly state that the reconstructed mass does not include SOA.

The Environmental Protection Department (?) is differently named in the reference list.

Kirchtetter et al. is misplaced in the alphabetical reference list.
* * *

---

## Author Comment (AC1) · 14 Sep 2017

We thank the referee for his/her time to provide us with extensive and valuable input. Please find below our responses to the raised comments, questions and suggestions. In the following, raised **comments / suggestions are in red** and respective **responses in green**, while **alterations to the manuscript text are indicated in blue.**

**General Comment**

This paper reports measurements of carbonaceous particles at a busy street canyon. The authors focus on primary hydrocarbon like (HOA) and elemental carbon (EC) fraction of the aerosol. They combine the concentration data with engine-type specific vehicle counts to obtain the contribution of diesel, gasoline, and LPG vehicles. The changes in emission regulations and the rapid development in engine and aftertreatment techniques make efforts such as this necessary in evaluating the effect of the policies and development on the air quality. The measurement site is ideal for this kind of study because of the high contribution of traffic and the local measures taken for decreasing the traffic emissions. The paper brings new information on an important question and merits publication, but there are some issues that should be treasted, as discussed below.

I understand that there are so many aspect on carbonaceous aerosols that the discussion needs to be limited somehow. However, I find the approach especially in the introduction too limited to the Hong Kong primary aerosol case. The introduction should shortly discuss the relative importance of the primary and secondary aerosol in urban settings and not just mention the secondary aspect in the last sentence of the paper. The authors should make a case why it is still important to study the primary components. In presenting and discussing the results, the authors should clearly, preferably in a table, report the primary and overall concentrations. The effect of renewing fuels and vehicle fleet on the urban concentrations has been studied also outside Hong Kong. These studies should be discussed, starting with the Harrison and Beddows, Nature 2017 paper. There are recent papers on the same measurement cite, partly by the same authors. The authors should clearly state what is new here, especially compared to the Lee at al. 2015 JGRA and Huang et al ACP 2014 papers. There is an important methodological difference to the Huang paper in treating the traffic related OC. The difference in methodology and results, as well as the meaning of those should be discussed more explicitly. Finally, the methodology and the analyses appear solid but they are treated so concisely that the reader needs to deduce what data was used. The methods are partly the same as used in the papers mentioned above, but this paper should also be readable separately. As this media allows some volume, the methodology should be described in more detail.

We have substantially revised the introductory part (→1) to reflect a more general discussion on the relevance and impacts of traffic emissions on primary and secondary aerosol pollution in urban environments. Due to the extent of the changes, the introductory section in its entirety is appended to the bottom of this document.

A clarifying sentence (→2) has been added to describe the differences to previous work.

The comparison of the AMS measurements to those using the EC/OC method has been extended (→ 3).

The methodology section has been revised to include all data analysis and processing steps that are relevant to the discussion of this work (→ 4) and an additional table is provided to give an overview of measured $PM_1$ and $PM_{2.5}$ mass concentrations and the traffic-related components as reported in this study (→ 5).

1 […]

*-- See end of document for revised introductory section --*

1 […]

2 […]

[revised manuscript text omitted]

5 […]

**Specific Comments**

**Abstract:**

**Comment** The abstract should highlight the new findings of this paper and report them quantitatively.

**Response** The abstract has been modified to include more of our quantitative results.

**Alteration** […]On an average per-vehicle-basis, gasoline vehicles emitted 75% and 93% more organics than diesel and LPG vehicles respectively, while EC emissions from diesel vehicles were 45% higher than those from gasoline vehicles. LPG vehicles showed no appreciable contributions to EC and thus overall represented a small contributor to traffic-related primary ambient PM$_1$ despite their high abundance in the traffic mix (~30%). Total carbonaceous particle mass contributions to ambient PM$_1$ from diesel engines were only marginally higher (~4%) than those from gasoline engines, which is likely an effect of recently introduced control strategies targeted at commercial vehicles and buses. Overall, gasoline vehicles contributed 1.2 µg/m³ of EC and 1.1 µg/m³ of organics, LPG vehicles 0.6 µg/m³ of organics and diesel vehicles 2.0 µg/m³ of EC and 0.7 µg/m³ of organics to ambient carbonaceous PM$_1$. […]

**Methodology:**

**Comment** The methodology part should be rewritten to work on its own for this paper. Reading just this paper, it remains unclear what instrument data was used for, e.g., the engine specific contributions. Was this done on hourly basis as is indirectly indicated on the vehicle flow chapter? Obviously AMS data was used for the HOA, but what about EC: Aethalometer or the hourly EC measurement? And how did the latter two compare?
Huang et al., 2014 should be cited already in the methodology part for the hourly EC/OC. An obvious measurement missing from the study is CO2. This inhibits the calculation of emission factors and should be clearly stated.

**Response** We have revised the methodology part to include more information on the data acquisition and treatment which are relevant to this work. As the reviewer correctly points out, CO$_2$ measurements were indeed not available thus preventing emission factor calculations, and we include a statement

to emphasize this circumstance. We provide an estimated per-vehicle contribution for each engine-type category in Figure 3b – while this was previously based on the vehicle numbers from the counting days we have scaled the contributions by the expected total daily vehicle number (average annual daily traffic – AADT) in the revised manuscript.

We mention the aethalometer in the methodology as it was integral to the instrumental setup used during the campaign, but was not operational continuously. There was significant overlap between BC and EC measurements in later parts of the sampling campaign (June-July) with good temporal agreement between the ECOC analyzer and the aethalometer (see below). We note a significant positive bias in the BC mass measurements (intercept of 1.7 µg/m$^3$) which may be caused by filter loading effects (Virkkula et al., 2007) or the presence of other light absorbing species, e.g. brown carbon (BrC) (Olson et al., 2015;Lin et al., 2015;Zhang et al., 2013). Due to these uncertainties and the lack of BC measurement data during the captioned time period covered in this work (late May 2013), we only discuss the hourly EC measurements.

[Figure]

**Figure R1.** Scatter plot of hourly BC (aethalometer) and hourly EC (ECOC analyzer) measurements (*open circles*) with orthogonal distance regression (*solid line*) between June 1 and July 19, 2013 at the Mong Kok site

**Alteration** […] Elemental and organic carbon (EC, OC) were measured (semi-continuous thermo-optical ECOC analyzer RT-3131, Sunset Inc., USA) at the AQMS at 1h time resolution and at similar sampling height (~2m) employing a modified National Institute for Occupational Safety and Health (NIOSH) method 5040 protocol which has been described in a previous study (Huang et al., 2014). Concentrations of EC and OC in PM$_1$ were estimated by converting the raw PM$_{2.5}$ concentrations using a PM$_1$ to PM$_{2.5}$ ratio of 0.8 representing the overall average of PM$_1$ to PM$_{2.5}$ ratios from previous roadside sampling studies (0.75-0.85) in Hong Kong (Cheng et al., 2006;Lee et al., 2006). AMS data were treated according to general AMS data treatment principles (DeCarlo et al., 2006;Jimenez et al., 2003) with standard software packages (SQUIRREL v1.53G, PIKA v1.12G). Analysis of the unit-mass resolution mass spectra yielded non-refractory submicron particle species concentrations of major inorganic constituents (SO4, NO3, NH4, Chl) and total organics at a base time resolution of 10 min. Positive Matrix Factorization (PMF) was used to deconvolute high-resolution organic mass spectra acquired at 10 min time resolution following recommended PMF guidelines for AMS data (Zhang et al., 2011) with the AMS PMF analysis toolkit (Ulbrich et al., 2009). Six organic aerosol (OA) factors were identified encompassing three secondary organic aerosol (SOA) and three primary organic aerosol (POA) factors of which one was attributed to traffic emissions (hydrocarbon-like organic aerosol, HOA(Zhang et al., 2005)) and two to cooking activities. OA factor concentrations were established using the fractional factor contributions from the high-resolution PMF analysis and the total submicron organics concentrations from the unit-mass resolution analysis. Further specific details on instrument parameters, data analysis and an overview of the general characteristics of submicron particulate matter from the same measurement campaign can be found in a previous publication (Lee et al., 2015) and are not directly relevant to this work. […]

[…] Vehicle emission factors are typically based on the concentration of emitted species per unit of consumed fuel (derived from $CO_2$ measurements) or per driven distance. As concurrent $CO_2$ measurement data were not available, emission factors were not evaluated in this study. Instead, the

relative emission behavior of different vehicle groups and their impacts on ambient PM concentrations were approximated by normalizing the resolved engine-specific carbonaceous primary aerosol concentrations by the total number of vehicles in each category (Figure 3b). […]

**Comment** Vehicle identification method to gain engine-type specific emissions seems to have been used at least in tunnels, although not necessarily documented very well. The authors should cite those, such as Cui et al., STOTEN 2016. They might also discuss the individual-vehicle specific approach of Wang et al., AtmEnv 2016.

**Response** We have included said references in the revised manuscript.

**Alteration** […] Details on traffic composition are usually obtained by concurrent video image recordings and subsequent manual assessment of vehicle numbers and identification of vehicle types with classifications e.g. by vehicle function (Wang et al., 2016) or broader groups such as fuel types (Cui et al., 2016). In this work, vehicle data were obtained from the Hong Kong government, which conducted a counting exercise during the sampling period over three weekdays (noon 28 May 2013 to noon 31 May 2013, Tue to Fri) using automated license plate recognition (ALPR) with four infrared cameras on-site. […] Non-personalized data on registered vehicles were obtained, including license class (*vehicle type*), year of manufacture (*vehicle age*) and engine type (*gasoline, diesel, LPG, others*), thus providing a direct linkage to actual vehicle inventory data and circumventing the need for manual assessments. […]

**Results:**

**Comment** The paper reports the contributions of the vehicle types, but also PM2.5, PM1, EC and OC concentrations should be given, maybe in a table gathering the results. Those reported earlier with a citation. At least Zhang et al., ACP 2005 should be cited for using HOA as a surrogate for combustion POA. In discussing the seasonal variations and gaseous/particulate phase partitioning, Robinson et al., 2007 should be cited, maybe together with comparison to some other studies, such as Budisulistrioni et al., ACP 2016; Huffmann et al., ACP 2009.

The paper reports exceptionally high EC fractions (EC/HOA ratios) for gasoline vehicles. This is an interesting and potentially important finding, but also subject to controversy. While the site-specific driving patterns could explain some of this, other aspects should also be discussed. A high fraction of gasoline direct injection (GDI, DISI) vehicles could possibly affect this, as they have been found to exhibit high primary EC fractions (e.g. Karjalainen et al., ACP 2016; Fushini et al., AtmEnv 2016). On the other hand, the HOA concentration of this study comes from AMS, while the EC does not. The low sampling efficiency of the AMS for sub-50 nm particles could cause low HOA concentration as a measurement artefact, especially if a high mass fraction of the POA is within the nucleation mode.

**Response** A table has been included *(see reply to general comment above: →5)*.

We have also added the suggested references in the revised manuscript. We further mention the possibility of GDI engines contributing to the high observed EC fraction. The vehicle data available in this study however are not specific enough to deduce the proportion of GDI vehicles among gasoline vehicles in this study.

Another paragraph has been added to discuss the portioning/semi-volatility of HOA and related studies (see below).

We also further stress the limitations of the AMS inlet in sampling nucleation mode particles. While this is relevant in terms of particle number concentration, the mass fraction of nucleation mode particles is low and we therefore believe that AMS inlet losses should have limited effects on the measured mass concentrations.

**Alteration** […] The semi-volatility of HOA in this study is inferred from the observed trend in ambient mass concentrations and affirms more recent studies on the partitioning behavior of POA components. Evaporation of semi-volatile components upon dilution decrease ambient POA concentrations considerably and are expected to vary with temperature (Robinson et al., 2007) as is the case in this study. Measurements of thermally denuded particles of primary origin by TD-AMS indicate that POA components can exhibit properties of semi-volatility on a scale similar to SOA in urban environments (Huffman et al., 2009a;Huffman et al., 2009b), and more recent studies suggest that HOA possesses a relatively wide volatility distribution with up to 50-60% of its mass consisting of semi volatile organic compounds (Cappa and Jimenez, 2010;Paciga et al., 2016). As noted earlier,

in this study the seasonal decrease in submicron particle-phase organic mass concentrations (HOA) by 40% was associated with a substantial increase in mean temperature by 7ºC between spring (23ºC ± 3ºC) and summer (30ºC ± 2ºC). […]

[…] Dynamometer studies mostly reported EC/OC values for gasoline powered vehicles < 1 (Alves et al., 2015;Geller et al., 2006) indicating overall a greater importance of organics in particulate emissions. The relatively high fraction of EC in gasoline related PM in our work likely stems from a combination of factors. HOA mass concentrations are derived from AMS measurements which are subject to limitations of inlet lens transmission, i.e. a standard lens as used in this study is expected to transmit efficiently (~100%) between 90nm and 700nm $D_{va}$ (Williams et al., 2013), and thus may cause a low bias in measured ambient HOA concentrations. However, these inlet losses are expected to only have limited effects on total measured submicron particle mass concentrations as mass- and volume-based particle concentrations from vehicle exhaust typically peak in the region of ~ 100 – 300nm (Fushimi et al., 2016;Ban-Weiss et al., 2010). The observed seasonal variability of particle-phase organic concentrations due to partitioning of semi-volatile components into the gas-phase would also decrease particle-phase HOA concentrations in warmer seasons, thus elevating the observed ambient EC/HOA ratio with our measurements taking place in summer. While this would affect all engine types, the extent may vary with the distribution of semi-volatile species in the respective tailpipe emissions of different vehicle groups. Due to the characteristics of the sampling site (inner-city urban road traffic and proximity to both a road junction and pedestrian crossing) sampled emissions include a considerable fraction of variable and higher engine loads during the acceleration phase whereas road tunnel environments are characterized by largely constant engine loads and traveling speeds. As gasoline vehicles are typically not equipped with particle filters, they have been shown to emit significant amounts of EC under unstable engine loads (Karjalainen et al., 2014). It has also been noted that changes in engine technology, i.e. the move from port fuel injection (PFI) to gasoline direct injection (GDI), may shift gasoline vehicle exhaust characteristics in favor of elemental carbon. Higher particulate matter mass emissions of GDI vehicles compared to PFI vehicles by a factor of 2 have been reported, which were mainly due to enhancements in EC emissions (Saliba et al., 2017). Similar observations were made in comparisons of PFI and direct injection spark ignition (DISI – a derivative of GDI) vehicles over both cold and hot-start conditions with higher total carbon (TC) emissions and higher EC/TC ratios for the DISI vehicles (Fushimi et al., 2016). At the same time, various control schemes targeting diesel vehicle emissions have been introduced in recent years. In Hong Kong these included […]

**Details:**

**Comment**    In section 3.3.2. it would be good to explicitly state that the reconstructed mass does not include SOA. The Environmental Protection Department (?) is differently named in the reference list. Kirchtetter et al. is misplaced in the alphabetical reference list.

**Response**    We have included a statement on the reconstructed mass as recommended. The reference list has been corrected as well.

**Alteration**    […] The time series of measured and reconstructed HOA and EC concentrations are depicted in Figure S5a and S5c in the Supporting Material respectively, representing the sum of primary carbonaceous particulate compounds and not including SOA species that may be formed through subsequent atmospheric processing of gas- and particle-phase species. […]

**Changes to main text**

**1.1. Particulate matter from motor vehicles in urban areas**

[revised manuscript text omitted]

---

## Referee Comment (RC2) · Anonymous Referee #3 · 27 Sep 2017

This paper presents measurement of ambient carbonaceous submicron particulate matter in a roadside site in urban Hong Kong. Multilinear regression analysis was performed with the observed HOA and EC concentrations and real traffic data to interpret the contribution of different vehicles to the carbonaceous aerosols. The results would be helpful to the local authority on traffic emission controls and be of interest to the community. The writing and construction of the paper need to be further improved before considering for acceptance, and the extremely long paragraph and sentences could confuse the reader and make the meaning ambiguous. Some parts in the introduction, methodology, and discussions also require more clarification and refinement.

**General comments:**

There are always long paragraphs and long sentences without appropriate breaks, and the lack of organization confuse the reader and diminish the intent of the discussions. For example, Line 81 to 112, single long paragraph for Section 3.1, line 279 to 317, and also some extremely long sentences in many places in the text.

Introduction. Since the present study made substantial analysis and discussion of the primary emission of HOA from different vehicles, the author needs to provide some review of the finding and current understanding of HOA from traffic emission in the introduction section, to better present what is new and significant in the present study.

Methodology. More details on the multilinear regression would be necessary, including inputs and outputs of the regression, any assumptions were made in the analysis, and the uncertainties raised by different factors, three-point box smoothing, constrained intercept, etc. For the sake of clarity, this information should be included to show the validity of the regression and analysis after that.

Results. Line 161-170, Line 182-186, and Line 189-193. The general descriptions of HOA during the campaign, relation with other species (e.g., NOx), the EC/HOA ratios, and the Sunday reductions of HOA and EC, all have been previously presented in the published paper by the same author (Lee et al., 2015). Thus it is suggested to condense the already published results and try to focus on what is new from the analysis in the present study.

Figure 1c, the author has already reported similar results from the same campaign in his previously published papers, i.e., Figure 4c in Lee et al., 2015, except that the current one uses EC in PM1 with an empirical ratio of 0.8 from PM2.5. It is also surprised that the same dataset for HOA gave different results between Lee et al., 2015 and the present study, see below comparison figures. The author needs to clarify this, otherwise, there are reasons to doubt the validity of the data and analysis in the present study.

[Figure]

Same problem for Figure 1a, the diurnal variations of EC and HOA in the summer campaign had also been published in the previous paper, i.e., Lee et al., 2015, as a supplement figure, Figure S5. Also, the Figure1Sb in this manuscript is also similar to Figure 4d in Lee et al., 2015.

Section 3.3.2 and Conclusion. The discussion on the control strategies did not show too many links with the results obtained in the present study, some of the statements seem too speculated without direct results or evidence to support, e.g., Line 332-335. I would suggest the author perform further analysis with other related species and data, and emphasize the indications from the results, and highlight what is new regarding findings in the present work in comparison to previous studies conducted in other places in the world.

**Specific comments:**

Line 34-39. The author stated that the previous investigations typically rely on source inventories with models, then how about the previous field measurements at the roadside environment? It is better to perform more comprehensive review regarding similar field measurements.

Line 40. It is not clear how different the approach in the present study compared to previous studies in the literature. The author should review the literature methods first and then can come up with the statement that the approach here is different and better.

Line 40-47. I would suggest the author re-locate the position of this paragraph to the place more fits the content, for example, the end of the Introduction section, where the description seems have some connection to this paragraph.

Line 63 to 67. Was the data measured in the present study also affected by the non-local pollution sources? Then how to differentiate the effects of local emission from non-local influence? Any

assumptions were made, and any uncertainties would be raised? The author should clarify this in the results and discussion section.

Line 61-67 and Line 75-80. The author listed many results from previous studies, however, the simple enumeration without refinement makes the descriptions confused. Another example can be found in Line 84-86, it is not clear about the purpose to mention the contribution to ozone formation, since nothing was discussed in this manuscript regarding the ozone issues.

Line 93-94, How about the share of OC in diesel emissions, since the emission factors of OC is about 8 times higher than that for non-diesel vehicles?

Line 97-99. Many studies reviewed in this section were conducted around or after 2010, e.g., Ning et al., 2011; Ho et al., 2013; Huang et al., 2014; Cheng et al., 2010; Yuan et al., 2013; Sun et al., 2016; Lee et al., 2015, etc. The author needs to clarify more on why '*they are unlikely to reflect the contemporary... over the last 15 years'* and the advantages of the present study to make progress on this issue.

Line 185-186. Did the concurrent measurement of hydrocarbon in the gas-phase show any pieces of evidence to support the hypothesis of more partitioning of HOA in gas-phase in summer?

Line 206-208. The purpose and links of these two sentences with the following discussion are not clear. The number of counted vehicles during the three-day counting exercise should be more useful here.

Section 3.3.1. The discussion of the selection of two-factor or three-factor models here seems clogged and could be largely condensed and refined. Also, as mentioned in methodology comments, more details on the validity and uncertainties of the analysis should be clarified.

Line 255-257. It is better to discuss the figure in the main text, i.e., Figure 3, and use the supporting figure as a supplementary discussion, otherwise, it makes the reader confused about the relationship and difference between Figure S5 and Figure 3.

Line 267-270, The multilinear regression is a statistical analysis that may not necessarily represent good physical meanings, it is necessary to compare the regression resolved emission factors with previous laboratory or field measured emission factors in Hong Kong or other regions, to validate the regression results.

Line 270-272, it is hard to understand what the author wants to interpret by only reading the text.

Please give a sense of the uncertainty of the obtained values, providing error bars in Figure 3b and 3e, and the uncertainties should be at the least qualitatively noted in the main text.

Line 290-291. It is not clear how the high fraction of EC in gasoline engine was related to the explanations here. More clarification is required to support the author's statement.

Line 295-300. Most of the previous studies reported higher particles for diesel vehicles compared to gasoline, is there any possibility that the different result in the present study was artifacts resulting from the statistics analysis lacking real physical meaning? Any other studies of direct emission measurement to support the similar low particles from diesel vehicles with DPF as the gasoline vehicles?

Line 313-317. What do these numbers mean and how can be linked to the results presented above? More discussions are needed here.

**Technical corrections:**

Line 68-71, grammar issues in the long sentences. Please rewrite it.

Line 76, Where was the open roadside located, the same site as the present study?

Line 81, which contributions did you refer to?

Line 99-103, reference needs to be provided

Line 175. 'was is' should be 'was'.

Line 209, the Figure S2a citing here is not the correct figure.

Line 286, uncompleted sentence.

---

## Author Comment (AC2) · 16 Oct 2017

We thank the referee for his/her time to provide us with extensive and valuable input. Please find below our responses to the raised comments, questions and suggestions. In the following, raised **comments / suggestions are in red** and respective **responses in green**, while **alterations to the manuscript text are indicated in blue.**

**General Comment**

This paper presents measurement of ambient carbonaceous submicron particulate matter in a roadside site in urban Hong Kong. Multilinear regression analysis was performed with the observed HOA and EC concentrations and real traffic data to interpret the contribution of different vehicles to the carbonaceous aerosols. The results would be helpful to the local authority on traffic emission controls and be of interest to the community. The writing and construction of the paper need to be further improved before considering for acceptance, and the extremely long paragraph and sentences could confuse the reader and make the meaning ambiguous. Some parts in the introduction, methodology, and discussions also require more clarification and refinement.

There are always long paragraphs and long sentences without appropriate breaks, and the lack of organization confuse the reader and diminish the intent of the discussions. For example, Line 81 to 112, single long paragraph for Section 3.1, line 279 to 317, and also some extremely long sentences in many places in the text.

We agree that there are occasional long sentences but we do not consider these to be too detrimental to the overall readability. We generally insert paragraph breaks if there is a thematic change, and keep contextually-related parts arranged together. In the revised manuscript, we have split longer paragraphs wherever possible to reduce the amount of long sections as suggested.

*-- Insertion of paragraph breaks throughout the main manuscript --*

Introduction. Since the present study made substantial analysis and discussion of the primary emission of HOA from different vehicles, the author needs to provide some review of the finding and current understanding of HOA from traffic emission in the introduction section, to better present what is new and significant in the present study.

We have substantially revised the introductory part to reflect a more general discussion on the relevance and impacts of traffic emissions on primary and secondary aerosol pollution in urban environments and the aims of our work. Due to the extent of the changes, the introductory section in its entirety is appended at the bottom of this document.

*-- See end of document for revised introductory section --*

Methodology. More details on the multilinear regression would be necessary, including inputs and outputs of the regression, any assumptions were made in the analysis, and the uncertainties raised by different factors, three-point box smoothing, constrained intercept, etc. For the sake of clarity, this information should be included to show the validity of the regression and analysis after that.

The multiple linear regression (MLR) is discussed in more detail in the corresponding result/discussion section (3.3.1.), since the immediately preceding discussion on the characteristics and variabilities in the measured carbonaceous species and traffic composition are critical in setting the context for the MLR analysis.

While we mentioned input and output information within the discussion, we more explicitly state these information at the beginning of section 3.3.1 in the revised manuscript.

Smoothing was performed due to the brevity of the data set (72h), where individual concentration or vehicle count spikes would have skewing effects on the regression analysis. We opted for box smoothing over three points as described in the methodology section to retain the general trends as close to the original data series as possible.

The choice of a zero intercept is based on the following considerations:

1.  HOA and EC are predominantly local pollutants originating from the direct vicinity of the sampling site. A further significant source of EC (and HOA) could be shipping, however, the nearest coastline is at a considerable distance (~ 2km) and shielded by complex, dense and tall urban building geometry. We consider a persistent influence from shipping emissions on ground-level pollution at this measurement site as unlikely.

2.  Organic species from farther sources (other city areas or regional pollution) are very likely to have undergone some form of atmospheric oxidation and would be accounted for in PMF-resolved oxygenated species (SOA – secondary organic aerosol).

3.  Traffic in the vicinity of the sampling site is largely homogenous (dense inner city traffic) and short-range transport through the street canyons should have little effect on the measured composition and concentration of carbonaceous $PM_1$ species.

4.  With a constant (non-zero) intercept, we would physically assume a constant background level (of either HOA or EC), regardless of changes in surface wind, air mass influence or possible diurnal changes in (background) source strength, which would not be sufficiently realistic.

We have expanded the discussion on the considerations for the MLR analysis in the revised manuscript.

*Section 2 (Methodology)*

[…] Measured HOA and EC during the traffic counting period were decomposed by multilinear regression (MLR) based on the time series of these engine count data, i.e. the hourly time series of HOA and EC were regarded as functions of the count of diesel, gasoline and LPG powered vehicles representing the independent regression variables. To reduce the skewing impact of scattered spikes in vehicle count number, HOA and EC concentrations, the time series

were subjected to three-point box smoothing prior to regression analysis. Multiple linear regression was performed for two and three factor solutions and with a constrained zero intercept. A non-zero intercept may not be physically meaningful as it requires the assumption of a constant background level of HOA and EC regardless of changes in surface wind, air mass influence or diurnal changes in background source strength. Contrarily, traffic is largely homogenous in the vicinity of the sampling site (dense inner city traffic) and other possible combustion sources (i.e. shipping) were sufficiently removed (~2km of straight-line distance to the nearest coastline) and shielded (complex and tall urban geometry) from the ground-level sampling site, i.e. impacts of transport on measured traffic-related carbonaceous constituents can be assumed to be of minor influence. MLR solutions were assessed per statistical significance of resolved regression coefficients, adjusted $R^2$ of the regression and the distribution of residuals. Further details are discussed in the corresponding discussion section 3.3.1. […]

*Section 3.3.1 (Results & Discussion)*
[…] Utilizing the detailed information on average daily traffic composition at the measurement site from the counting exercise, contributions of exhaust from different engine types to overall ambient HOA and EC concentrations can be evaluated. Multiple linear regression was carried out with the concentration time series of HOA and EC during the traffic counting period as the dependent variables, the pooled sum of counted diesel, gasoline and LPG vehicles as the independent variables, and a constrained zero intercept, assuming negligible transport of HOA and EC (see section 2). Two factor solutions (diesel + gasoline), as well as three factor solutions (diesel + gasoline + LPG), were considered, as LPG vehicles are expected to contribute less to particle-phase and more to gas-phase emissions (Faiz et al., 1996). The key statistical output parameters (adjusted $R^2$, regression coefficients, p-values) of the multiple linear regression analysis for HOA and EC for two and three factor models are presented in Table S1 in the Supporting Material. […]

Results. Line 161-170, Line 182-186, and Line 189-193. The general descriptions of HOA during the campaign, relation with other species (e.g., NOx), the EC/HOA ratios, and the Sunday reductions of HOA and EC, all have been previously presented in the published paper by the same author (Lee et al., 2015). Thus it is suggested to condense the already published results and try to focus on what is new from the analysis in the present study.
While we agree that parts of the methodology and underlying data are similar as in the mentioned overview paper, this manuscript is not a companion paper and should remain readable as a separate analysis. The concerned parts of the analysis are essential in providing the context for the discussion of the work of this study and should therefore be retained.

Figure 1c, the author has already reported similar results from the same campaign in his previously published papers, i.e., Figure 4c in Lee et al., 2015, except that the current one uses EC in PM1 with an empirical ratio of 0.8 from PM2.5. It is also surprised that the same dataset for HOA gave different results between Lee et al., 2015 and the present study, see below comparison figures. The author needs to clarify this, otherwise, there are reasons to doubt the validity of the data and analysis in the present study.
While the base data are the same for both studies, we have employed a different analysis procedure: in the overview paper (Lee et al., 2015) the entire available dataset of EC and HOA data was used in the calculation of the Weekday-Sunday difference.
As data gaps (e.g. due to instrument maintenance) may bias average concentrations - especially if diurnal variations are well pronounced as is the case for traffic-related pollutants at this sampling site - we excluded days where data availability was < 75% (i.e. more 6h of data missing) from both HOA and EC datasets in this work. This leads to slight differences in nominal concentrations (and thus relative reductions) between this work and the cited work but does not affect the overall conclusions in either study. We consider the method employed here an improvement in accuracy compared to the previous work and we will point this out more specifically in the revised manuscript for clarification.
[…] Both HOA and EC exhibited significant reductions in overall mass concentrations on Sundays between 14% and 27%, compared to the rest of the week (Monday to Saturday) as shown in Figure 1d. Days with less than 75% (>18h) data availability were excluded from both time series in the analysis to obviate bias in the daily averaging from their strong diurnal patterns, as opposed to the original data presented by Lee at al. (2015) where the entire unabridged time series were used. […]

Same problem for Figure 1a, the diurnal variations of EC and HOA in the summer campaign had also been published in the previous paper, i.e., Lee et al., 2015, as a supplement figure, Figure S5. Also, the Figure1Sb in this manuscript is also similar to Figure 4d in Lee et al., 2015.
The analysis of traffic related carbonaceous aerosol components in this work has been substantially expanded compared to the overview paper (Lee et al., 2015). As discussed above, to ensure readability as a standalone paper, we include previously analyzed data to provide sufficient background information required for the context of the discussions in this paper.

Section 3.3.2 and Conclusion. The discussion on the control strategies did not show too many links with the results obtained in the present study, some of the statements seem too speculated without direct results or evidence to support,

e.g., Line 332-335. I would suggest the author perform further analysis with other related species and data, and emphasize the indications from the results, and highlight what is new regarding findings in the present work in comparison to previous studies conducted in other places in the world.

A direct evaluation of control strategy impacts requires a long-term database of measurement data in terms of vehicle-related ambient species as well as detailed traffic count and composition data, which are unfortunately not available. The local specificity of both traffic characteristics and regulatory control strategies similarly limits direct inferences to be made from studies from other parts of the world.

Nonetheless, as presented in this manuscript, we examined findings from various recent studies on the emission characteristics of vehicles, including high EC fractions in GDI gasoline vehicles, efficient total PM reduction in diesel vehicles equipped with exhaust after treatment devices (such as DPF, EGR) as well as overall changes in ambient concentrations observed at the roadside in Hong Kong which altogether are consistent with the results obtained from our analysis of the measurement data. Apart from the mentioned species (VOCs, criteria gas pollutants, NR-PM$_1$, EC, OC, meteorological data), no other supporting data were available in the captioned time period and we note that "carbonaceous particulate matter" is the focal point of this manuscript.

**Specific Comments**

| | |
|---|---|
| **Comment** | Line 34-39. The author stated that the previous investigations typically rely on source inventories with models, then how about the previous field measurements at the roadside environment? It is better to perform more comprehensive review regarding similar field measurements. |
| **Response** | We have substantially revised the introductory part and include further references and discussions to both field and laboratory measurements. |
| **Alteration** | *-- See end of document for revised introductory section --* |
| **Comment** | Line 40. It is not clear how different the approach in the present study compared to previous studies in the literature. The author should review the literature methods first and then can come up with the statement that the approach here is different and better. |
| **Response** | As mentioned above, we have included further references and discussions to both field and laboratory measurements. The aim of this paper is to present results from one possible approach to investigate the role of different exhaust types and their influence on ambient species concentrations (as opposed to emission factor or exhaust characterization studies) utilizing more detailed traffic data. "Different" as used in our introductory section refers to the previous description of other commonly used methods and the evaluation of the performance of our method to other methods is beyond the scope of this manuscript. |
| **Alteration** | *-- See end of document for revised introductory section --* |
| **Comment** | Line 40-47. I would suggest the author re-locate the position of this paragraph to the place more fits the content, for example, the end of the Introduction section, where the description seems have some connection to this paragraph. |
| **Response** | We have revised the introductory part with added discussions on the role of motor-vehicle derived particulate matter, improving the contextual relationship of this paragraph to the introductory section. |
| **Alteration** | *-- See end of document for revised introductory section --* |
| **Comment** | Line 63 to 67. Was the data measured in the present study also affected by the non-local pollution sources? Then how to differentiate the effects of local emission from non-local influence? Anyassumptions were made, and any uncertainties would be raised? The author should clarify this in the results and discussion section. |
| **Response** | We refer to our response in the general comment section with regard to the methodology part. HOA is freshly emitted POA with corresponding mass spectral characteristics. There are no other significant sources of fossil fuel combustion in the area, except for shipping. As mentioned earlier, the location of the measurement site within a street canyon and radially surrounded by dense and high building structures make a consistent influence of shipping emissions on our ground-level measurements unlikely. The same applies to EC. Other non-local pollutants are likely to have undergone atmospheric processing during transport and would be resolved as other factors (e.g. SOA factors in the PMF analysis), which are not discussed in this work. |
| **Alteration** | N.A. |
| **Comment** | Line 61-67 and Line 75-80. The author listed many results from previous studies, however, the simple enumeration without refinement makes the descriptions confused. Another example can be found in Line 84-86, it is not clear about the purpose to mention the contribution to ozone formation, since nothing was discussed in this manuscript regarding the ozone issues. |
| **Response** | Studies on traffic emissions from the South East Asian region are relatively scarce and we therefore provide a more detailed overview of the research efforts in Hong Kong to provide a context for the reader to understand the location-specific circumstances, especially in contrast to Europe and the USA (lower building and population density) or the mainland of China (differences in vehicle composition/technology). While the ozone issue may not be directly related to this work, we included |

such references for a more complete overall picture.

We offer a more compact discussion of studies using similar methodologies, e.g. the ambient EC and OC measurement based studies. For gas-phase components and emission factor studies, methodologies differ widely across the reviewed papers and we therefore summarize their key findings on a more individualistic basis.

| | |
|---|---|
| **Alteration** | N.A. |
| **Comment** | Line 93-94, How about the share of OC in diesel emissions, since the emission factors of OC is about 8 times higher than that for non-diesel vehicles? |
| **Response** | The same study reported a 26% share of OC in the $PM_{2.5}$ emissions of diesel vehicles. |
| **Alteration** | N.A. |
| **Comment** | Line 97-99. Many studies reviewed in this section were conducted around or after 2010, e.g., Ning et al., 2011; Ho et al., 2013; Huang et al., 2014; Cheng et al., 2010; Yuan et al., 2013; Sun et al., 2016; Lee et al., 2015, etc. The author needs to clarify more on why *'they are unlikely to reflect the contemporary... over the last 15 years'* and the advantages of the present study to make progress on this issue. |
| **Response** | While it is correct that these studies were published in fairly recent years, most of the measurements reporting EC and OC concentrations or emissions explicitly in these studies were undertaken in the early 2000s (e.g. Cheng et al, 2010 reports on measurements from 2003 as mentioned in the same paragraph in the manuscript, while Yuan et. al., 2013 covered filter samples up to 2008 only as mentioned in the introductory section). Changes in traffic mix and engine technology / exhaust treatment since then are highly likely to have taken place and our more recent measurements would yield a more up-to-date evaluation of traffic-exhaust contributions to ambient PM. |
| **Alteration** | N.A. |
| **Comment** | Line 185-186. Did the concurrent measurement of hydrocarbon in the gas-phase show any pieces of evidence to support the hypothesis of more partitioning of HOA in gas-phase in summer? |
| **Response** | Only measurements of VOC compounds were available, which do not include higher molecular weight species that form part of the semi-volatile (SV) fraction of HOA. |
| **Alteration** | N.A. |
| **Comment** | Line 206-208. The purpose and links of these two sentences with the following discussion are not clear. The number of counted vehicles during the three-day counting exercise should be more useful here. |
| **Response** | As discussed in the methodology section, the traffic counting exercise did not continuously monitor all lanes of traffic at the site and therefore, we here elaborate on the relationship between the counted traffic numbers and the expected total number of vehicles from official government statistical data. |
| **Alteration** | N.A. |
| **Comment** | Section 3.3.1. The discussion of the selection of two-factor or three-factor models here seems clogged and could be largely condensed and refined. Also, as mentioned in methodology comments, more details on the validity and uncertainties of the analysis should be clarified. |
| **Response** | *See response to General Comments (Methodology).*

 We consider the placement of a more detailed discussion on the multiple linear regression at this place in the manuscript more appropriate as the preceding discussion on the trends in both carbonaceous primary PM and traffic composition are vital in providing the context for the discussion of the MLR analysis.

 Validity and uncertainties from the MLR analysis are discussed in response to a subsequent comment ($\rightarrow$ *see response to comment on Line 267-270 further below*). |
| **Alteration** | N.A. |
| **Comment** | Line 255-257. It is better to discuss the figure in the main text, i.e., Figure 3, and use the supporting figure as a supplementary discussion, otherwise, it makes the reader confused about the relationship and difference between Figure S5 and Figure 3. |
| **Response** | We have amended the wording and added a reference to Figure 3 in the said sentence to clarify the relationship between the main and supplementary figures. |
| **Alteration** | […] The time series of measured and reconstructed HOA and EC concentrations based on the regression coefficients are depicted individually in Figure S5a and S5c in the Supporting Material and in combination in Figure 3 to represent the sum of motor vehicle related primary carbonaceous particulate compounds […] |
| **Comment** | Line 267-270, The multilinear regression is a statistical analysis that may not necessarily represent good physical meanings, it is necessary to compare the regression resolved emission factors with previous laboratory or field measured emission factors in Hong Kong or other regions, to validate the regression results. |
| **Response** | The aim of this work is the evaluation of contributions of traffic emissions to ambient concentrations in contrast to emission factor studies, which are based on the composition and concentrations from the tailpipe. Comparability to emission factors is limited due to the lack of either $CO_2$ concentration & fuel consumption data (*concentration per fuel consumed*), or travelling distance of vehicles (*concentration* |

*per mileage*) as well as the influence of dispersion (*sampling at 3m of height, to the side of the road*) to determine emission factors from our measurement data.

We have demonstrated the considerations of our MLR analysis (statistical significance, traffic data, reference to previous studies) in section 3.1.1. validating its mathematical soundness.

While mathematically sound and physically meaningful solutions cannot automatically be equated, we have provided a comprehensive discussion of our results with both ambient and laboratory studies of more recent years in Section 3.2.2 (similar ambient studies, exhaust characterization studies, exhaust control devices). As they widely agree with the observations from our study - considering the specific characteristics of the vehicle mix and the sampling site environment – we are confident that our results are also physically meaningful.

| | |
|---|---|
| **Alteration** | N.A. |
| **Comment** | Line 270-272, it is hard to understand what the author wants to interpret by only reading the text. |
| **Response** | We have revised the wording for clarification. |
| **Alteration** | […] Each diesel and gasoline vehicles contributed 75-85% more carbonaceous $PM_1$ mass than LPG vehicles, which despite making up ~30% of total counted traffic accounted for less than 13% of traffic-related primary $PM_1$ (Figure 3e). […] |
| **Comment** | Please give a sense of the uncertainty of the obtained values, providing error bars in Figure 3b and 3e, and the uncertainties should be at the least qualitatively noted in the main text. |
| **Response** | Standard deviations of the fit parameters have been included in Table A1 in the Supporting Material and we include error bars in Figure 3b (main text) and standard deviations derived from the fit parameter uncertainties (Table S1, Supporting Material) in the pie charts of Figure 3e (main text) and Figure S5b,d (Supporting Material). |
| **Alteration** | |

[Figure]

**Figure 3.** […]

| | |
|---|---|
| **Comment** | Line 290-291. It is not clear how the high fraction of EC in gasoline engine was related to the explanations here. More clarification is required to support the author's statement. |
| **Response** | Lines 290-91 are to be seen in context with the following two sentences and associated references detailing how unstable engine loads (e.g. acceleration phase due to the sampling site location at a crossing / junction) lead to higher fractions of EC in gasoline vehicle exhaust compared to studies next to open (i.e. straight) roadways or in tunnels, where vehicles travel at relatively constant speed and thus at more constant engine loads. Further to this, recent studies also indicate that changes in vehicle technology (e.g. gasoline direct injection, GDI) are likely to shift tailpipe composition of gasoline vehicles more in favor of elemental carbon. We have added this discussion in the revised manuscript. |
| **Alteration** | […] It has also been noted that changes in engine technology, i.e. the move from port fuel injection (PFI) to gasoline direct injection (GDI), may shift gasoline vehicle exhaust characteristics in favor of elemental carbon. Higher particulate matter mass emissions of GDI vehicles compared to PFI vehicles by a factor of 2 have been reported, which were mainly due to enhancements in EC emissions (Saliba et al., 2017). Similar observations were made in comparisons of PFI and direct injection spark ignition (DISI – a derivative of GDI) vehicles over both cold and hot-start conditions with higher total carbon (TC) emissions and higher EC/TC ratios for the DISI vehicles (Fushimi et al., 2016). […] |

| | |
|---|---|
| **Comment** | Line 295-300. Most of the previous studies reported higher particles for diesel vehicles compared to gasoline, is there any possibility that the different result in the present study was artifacts resulting from the statistics analysis lacking real physical meaning? Any other studies of direct emission measurement to support the similar low particles from diesel vehicles with DPF as the gasoline vehicles? |
| **Response** | We would like to point out that the reported measurements in this study are in terms of particle mass which does not imply that diesel vehicles emitted a smaller number of submicron particles as this would depend on the characteristics of the particle size distributions of each engine type. |
| | Low particle mass emissions for diesel vehicles with fitted DPFs have been reported in previous studies (Li et al., 2014;Quiros et al., 2015;Mathis et al., 2005) where emission factors were either comparable (or lower) than those from port-fuel injection (PFI) gasoline vehicles. In contrast, gasoline direct injection (GDI) vehicle often exhibited larger PM emissions than both PFI gasoline and DPF-equipped diesel vehicles, as noted previously. |
| **Alteration** | N.A. |
| **Comment** | Line 313-317. What do these numbers mean and how can be linked to the results presented above? More discussions are needed here. |
| **Response** | As noted previously, advanced exhaust after treatment has been shown to greatly reduce particulate mass emissions in diesel vehicles. The vehicle data available in this study comprised information on the year of manufacture of the vehicle, but no details on the engine specifications (except for fuel type) or built-in exhaust after treatment devices. The requirement to comply with emission standards (e.g. Euro or LEV standards) have led to strict PM emission limits in newer vehicle models and are typically achieved through such exhaust after treatment devices. The vehicle manufacture year (compared to the introduction year of the respective Euro standard) can be regarded as a predictor for the fitting with advanced exhaust after treatment devices, given that most diesel vehicles in Hong Kong are imported from Europe, the US, Japan or Korea where manufacturers comply with such respective standards. |
| | We have modified the section to further stress its relationship with the discussion of our results. |
| **Alteration** | […]At the same time, various control schemes targeting diesel vehicle emissions have been introduced in recent years. In Hong Kong these included inter alia an incentive scheme […] |
| | […] New engine technologies for diesel vehicles, such as DPFs, have been shown to greatly reduce both EC and OC emissions (Alves et al., 2015) thus leading to overall little total particle mass emissions (Li et al., 2014;Quiros et al., 2015) partially due to shifts in the particle size distributions towards a greater fraction of particles in the ultrafine mode (Giechaskiel et al., 2012). The Euro III, IV and V standards for trucks and buses were introduced in late 2000, late 2005 and late 2008 respectively. While the year of manufacture does not directly infer compliance to a specific emission standard, an approximation of the fraction of vehicles fulfilling a certain standard can be made by assuming that vehicles produced between 2001 and 2005, between 2006 and 2008 and between 2009 and 2013 very likely comply with the Euro III, IV and V standards respectively. In this case, it is assumed that emission standards were immediately or had already been adopted in vehicle models in the corresponding year of manufacture. Figure S6 in the Supporting Material depicts the number and proportion of vehicles of certain years of manufacture and their assumed Euro standard. For goods vehicles, 52% of counted vehicles were built between 2005 and 2013 (i.e. likely fulfilling Euro IV and V), while for buses the proportion was slightly lower at 33%. With these two vehicle groups representing the bulk of diesel powered vehicles, an estimated 40% of diesel vehicles complied with Euro IV and Euro V standards during the time of our ambient measurements, further rationalizing the relatively low per-vehicle contribution of diesel vehicles to ambient exhaust-derived carbonaceous $PM_1$ in this study. [...] |

**Technical corrections:**

| | |
|---|---|
| **Comment** | Line 68-71, grammar issues in the long sentences. Please rewrite it. |
| **Response** | We could not identify any obvious grammatical mistake in the captioned sentence. However, we have split the sentence to improve its clarity in the revised manuscript. |
| **Alteration** | […] Generally, low overall organic carbon (OC) to elemental carbon (EC) ratios (0.6-0.8) which are typical for locations in direct proximity to primary combustion sources were observed. In comparison, samples from urban rooftop sites exhibited lower contributions of carbonaceous constituents (<50%) and were impacted more by oxidized secondary organic species with correspondingly higher overall OC/EC ratios ~1.9 (Louie et al., 2005;Cheng et al., 2010;Lee et al., 2006). […] |
| **Comment** | Line 76, Where was the open roadside located, the same site as the present study? |
| **Response** | The referenced studies were conducted at different locations, which also included the site of this present study. We note that open roadside here is used in contrast to "enclosed" tunnel studies which are also discussed in the same paragraph. |
| **Alteration** | N.A. |
| **Comment** | Line 81, which contributions did you refer to? |
| **Response** | We have revised the sentence for clarification. |
| **Alteration** | […] Contributions of traffic emissions to ambient gas-phase species were evaluated utilizing VOC canister samples collected in locations with different dominant-vehicle types in Hong Kong in 2003 (Ho et al., 2013). […] |
| **Comment** | Line 99-103, reference needs to be provided |
| **Response** | We have moved the captioned reference accordingly. |
| **Alteration** | […]A mobile measurement platform with an array of on-board PM and gas monitors was deployed in early 2012 for a more up-to-date characterization of fuel-based emission factors of PM, $NO_x$, and butane by chasing vehicle plumes on major roadways and highways (Ning et al., 2012) […] |
| **Comment** | Line 175. 'was is' should be 'was'. |
| **Response** | We have rectified this mistake in the revised manuscript. |
| **Alteration** | […] 90% of the daily HOA was accumulated between the start of the rush hour and midnight.[…] |
| **Comment** | Line 209, the Figure S2a citing here is not the correct figure. |
| **Response** | The Figure reference has been amended in the revised manuscript. |
| **Alteration** | […] The diurnal variation of the counted vehicle number, resolved by broad vehicle classes, and their varying contribution to the vehicle mix during the day are depicted in Figure B2c in the Supporting Material. […] |
| **Comment** | Line 286, uncompleted sentence. |
| **Response** | We have removed an erroneous "and" at the end of the captioned sentence. |

[revised manuscript text omitted]

---

## Author Response (AR2)

We thank the referee for his/her time to provide us with extensive and valuable input. Please find below our responses to the raised comments, questions and suggestions. In the following, raised **comments / suggestions are in red** and respective **responses in green**, while **alterations to the manuscript text are indicated in blue.**

**General Comment**

Overall the revised manuscript is much improved, and most of the comments have been addressed and alterations have been made.

However, I still think a large amount of duplication/reproduction of previously published results should be avoided in any new manuscript, even if the authors want to use the published results to support their further discussion or analysis. The relevant parts of the previously published results should be refined and concise in the present work, and the obvious duplication of general descriptions should be eliminated.

Therefore, I would suggest the authors condense the relevant parts and only retain the essential points that are related to the further analysis. The previously published figures are also suggested to be removed from the main manuscript, and only essential ones should be included in the supplementary.

Overall, the present manuscript is a significant extension based on data from our previous publication and we cannot completely eliminate a certain degree of overlap.

We have condensed descriptions in the methodology part relating to our previous work as far as possible. We believe that essential sections, e.g. sampling and data treatment details, should nonetheless remain for better contextual clarity.

We refer only briefly to the general characteristics of HOA and EC in this paper as an essential prerequisite for the understanding of the subsequent analyses. As the focus of this work is specifically on traffic-related PM (as opposed to our previous publication which presents a general campaign overview), we consider retaining these descriptions as beneficial to this manuscript as an entirely self-contained paper.

With regard to figures, the current revised manuscript contains a small number of subfigures which are of similar design as in our previous publication, but no duplicates. Due to differences in data analysis procedures (*noted in the main text:* data exclusions in the weekday analysis; conversion of $PM_{2.5}$ concentrations to $PM_1$) the depicted data are derived but not identical to our previous work, and should therefore not be removed.

--Deletion/shortening of sentences in Methodology (Sect. 2).--

In addition, for the multilinear regression, since pre-average to hourly means has been performed to all the input data including vehicle count numbers, HOA and EC concentrations, the three-point box smoothing could possibly introduce extra errors or uncertainties. The author is suggested to include the comparison of smoothing and non-smoothing results in supplementary and to discuss the possible uncertainties raised by the smoothing method.

We have included a clarifying paragraph in the main text, and an additional table (Table S2) and figure (Figure S5) each in the supplement relating to the smoothing of the time series.

In brief, smoothing improved the performance of the multiple linear regression with higher adjusted $R^2$ as well as lower average and summed residuals, and we therefore consider smoothing to reduce uncertainties which arise from the spikiness of the original data (particularly the raw hourly vehicle counts) as depicted in Figure S5.

[…] As discussed in the methodology section, three point box smoothing was applied to the vehicle counts and carbonaceous $PM_1$ species concentrations to reduce biasing of resolved regression parameters induced from spikes in the time series (Figure S5 in the Supplement). Comparing MLR results from the smoothed and raw time series data (Table S2 in the Supplement), impacts on the variable coefficients and their variability (standard deviations) were minor and factor statistical significance was not detrimentally affected. The smoothed data set overall yielded better results with higher adjusted $R^2$ as well as lower average and summed residual values. The smoothing process thus did not incur substantial additional uncertainties in the deconvolution of HOA and EC. […]

**Table S2** –Two factor (for EC) and three factor (for HOA) MLR regression statistics; Comparison between values obtained with raw vehicle counts & mass concentrations ("Raw") and smoothed (three point box smoothing; "Smth.") vehicle counts & mass concentrations

| Regression statistics | | HOA, 3 Fac Smth. | HOA, 3 Fac Raw | EC, 2 Fac Smth. | EC, 2 Fac Raw |
|---|---|---|---|---|---|
| $R^2_{adj}$ | Total | 0.90 | 0.87 | 0.92 | 0.87 |
| Residual | Ave. | 0.11 | 0.15 | 0.16 | 0.22 |
| Residual | Sum | 7.9 | 10.5 | 11.4 | 15.7 |
| P-Value | G-veh. | $5.63 \times 10^{-3}$ | $7.46 \times 10^{-3}$ | $1.51 \times 10^{-3}$ | $8.86 \times 10^{-4}$ |
| | D-veh. | $5.47 \times 10^{-3}$ | $4.69 \times 10^{-3}$ | $6.41 \times 10^{-8}$ | $3.61 \times 10^{-6}$ |
| | L-veh. | $4.86 \times 10^{-3}$ | $6.64 \times 10^{-3}$ | N.A. | N.A. |
| Var. coeff. | G-veh. | $1.10 \times 10^{-2}$ $\pm 3.86 \times 10^{-3}$ | $1.01 \times 10^{-2}$ $\pm 3.33 \times 10^{-3}$ | $1.12 \times 10^{-2}$ $\pm 3.40 \times 10^{-3}$ | $1.26 \times 10^{-2}$ $\pm 3.83 \times 10^{-3}$ |
| | D-veh. | $7.05 \times 10^{-3}$ $\pm 2.46 \times 10^{-3}$ | $7.46 \times 10^{-3}$ $\pm 2.80 \times 10^{-3}$ | $1.82 \times 10^{-2}$ $\pm 3.00 \times 10^{-3}$ | $1.72 \times 10^{-2}$ $\pm 3.26 \times 10^{-3}$ |
| | L-veh. | $6.17 \times 10^{-3}$ $\pm 3.08 \times 10^{-3}$ | $6.87 \times 10^{-3}$ $\pm 2.47 \times 10^{-3}$ | N.A. | N.A. |

[Figure]

**Figure S5.** Time series of raw vehicle count and raw EC and HOA mass concentrations (solid lines) and their counter parts after three point box smoothing (hashed lines)

[revised manuscript text omitted]